# Gold-nanofève surface-enhanced Raman spectroscopy visualizes hypotaurine as a robust anti-oxidant consumed in cancer survival

Megumi Shiota [1], Masayuki Naya[1], Takehiro Yamamoto [2], Takako Hishiki [2], Takeharu Tani[1], Hiroyuki Takahashi[3], Akiko Kubo[2], Daisuke Koike[4], Mai Itoh[2], Mitsuyo Ohmura[2], Yasuaki Kabe[2], Yuki Sugiura[2], Nobuyoshi Hiraoka [5], Takayuki Morikawa[6], Keiyo Takubo[6], Kentaro Suina[7], Hideaki Nagashima [7], Oltea Sampetrean [7], Osamu Nagano [7], Hideyuki Saya [7], Shogo Yamazoe[1], Hiroyuki Watanabe [1] & Makoto Suematsu [2]

Gold deposition with diagonal angle towards boehmite-based nanostructure creates random arrays of horse-bean-shaped nanostructures named gold-nanofève (GNF). GNF generates many electromagnetic hotspots as surface-enhanced Raman spectroscopy (SERS) excitation sources, and enables large-area visualization of molecular vibration fingerprints of metabolites in human cancer xenografts in livers of immunodeficient mice with sufficient sensitivity and uniformity. Differential screening of GNF-SERS signals in tumours and those in parenchyma demarcated tumour boundaries in liver tissues. Furthermore, GNF-SERS combined with quantum chemical calculation identified cysteine-derived glutathione and hypotaurine (HT) as tumour-dominant and parenchyma-dominant metabolites, respectively. CD44 knockdown in cancer diminished glutathione, but not HT in tumours. Mechanisms whereby tumours sustained HT under CD44-knockdown conditions include upregulation of PHGDH, PSAT1 and PSPH that drove glycolysis-dependent activation of serine/glycine-cleavage systems to provide one-methyl group for HT synthesis. HT was rapidly converted into taurine in cancer cells, suggesting that HT is a robust anti-oxidant for their survival under glutathione-suppressed conditions.

[1] Frontier Core-Technology Laboratories, Research & Development Management Headquarters, FUJIFILM Corporation, 577, Ushijima, Kaisei-machi, Ashigarakami-gun, Kanagawa 258-8577, Japan. [2] Department of Biochemistry, Keio University School of Medicine, Tokyo 160-8582, Japan. [3] Analysis Technology Center, Research & Development Management Headquarters, FUJIFILM Corporation, 577, Ushijima, Kaisei-machi, Ashigarakami-gun, Kanagawa 258-8577, Japan. [4] Functional Materials R&D Support Department, Research & Development Process Services Division, FUJIFILM Business Expert Corporation, 577, Ushijima, Kaisei-machi, Ashigarakami-gun, Kanagawa 258-8577, Japan. [5] Pathology Division, National Cancer Center Research Institute, Tokyo 104-0045, Japan. [6] Department of Stem Cell Biology, Research Institute, National Center for Global Health and Medicine, Tokyo 162-8655, Japan. [7] Division of Gene Regulation, Institute for Advanced Medical Research, Keio University School of Medicine, Tokyo 160-8582, Japan. Correspondence and requests for materials should be addressed to M.N. (email: masayuki.naya@fujifilm.com) or to M.S. (email: gasbiology@keio.jp)

New technology that visualizes multiple metabolites in a large surface area of tissues, but does not require labelling or staining, could improve pathological diagnosis. Raman scattering is a powerful analytical technique for objective, label-free tissue diagnosis[1-3]. Unlike conventional Raman spectroscopy, surface-enhanced Raman spectroscopy (SERS) yields highly sensitive signals of vibrational fingerprints of metabolites in the presence of reporter metals such as gold (Au) or silver[4,5]. Although SERS technology has been successful in biosensing and biomedical contexts: previous reports described the imaging of molecules or drug compounds at the single cell levels or in ex-vivo biofluid samples[6-8]. SERS imaging of large areas of tissues, with high sensitivity, spatial resolution and reproducibility, has been unavailable. Challenges in developing devices for practical use include the design of highly sensitive SERS—active substrates that guarantee spatial uniformity of hotspots, to amplify SERS signals by enhancing local electromagnetic fields[2]. Other difficulties arise from the technical principle underlying SERS, which detects multiple molecular vibration modes as fingerprints when identifying metabolites[1]. In tissues, metabolites yield numerous

SERS signals and complicate identification of their discernible molecular entities in individual Raman peaks. However, SERS imaging is a comprehensive technique for visualizing metabolic profiles at multiple different wave numbers, and requires only a modest laser excitation, which minimizes artificial oxidation of metabolites. These characteristics are useful in accurately determining biomarker metabolites in diagnosing cancer. Development of a highly sensitive SERS imaging modality that can cover a large area aids our understanding of the metabolic interplay between cancer cells and surrounding tissues and assists in objective and automated identification of tumour (T) boundaries in tissues[9,10].

The current SERS technique enables precise control of Au nanostructures of ideal geometry and produces gold nanofève substrates (GNF), which are named after the horse bean-shaped Au nanoparticles. GNF provides SERS excitation sources, thereby yielding numerous strong signals derived from metabolites in livers bearing metastatic cancer xenografts. Non-target differentiation of such signals in T and parenchyma (P) helped identify T boundaries in frozen tissue sections by means of automated

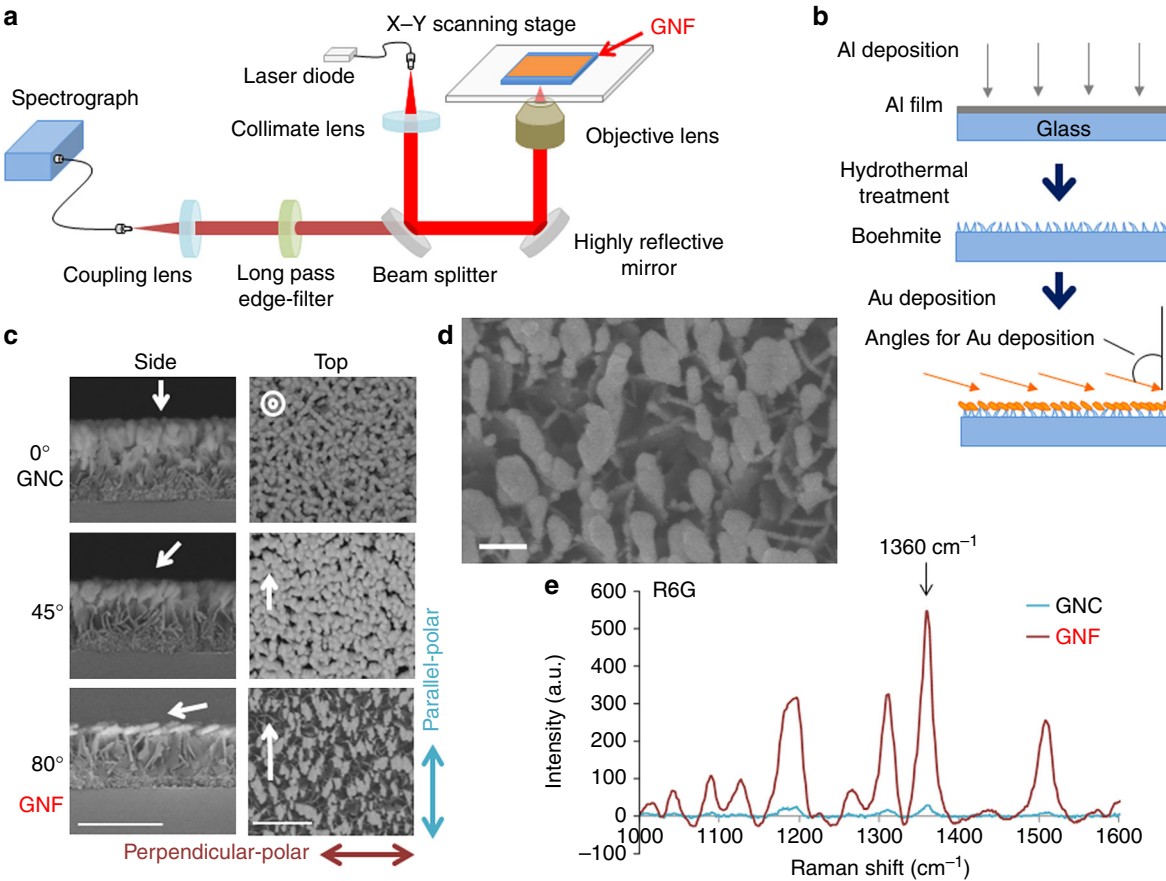

**Fig. 1** Characterization of a random array of the gold-nanofève (GNF) substrate for SERS imaging. **a** Schematic diagram of SERS imaging spectrometry system. Note that the system is based on the inverted-type microscope in which the GNF substrate is optically transparent, and allows us to visualize SERS signals from the bottom of the substrate without being interfered with the thickness of tissue slices on the microscopic stage. **b** Nano-fabrication of GNF substrates. Deposition of aluminum (Al) onto the surface of a film layer is followed by immersion of the Al-coated film into boiling water as a hydrothermal treatment that forms thin, sharp-edged leaf-like superstructure of boehmite, as shown previously from our laboratories[15]. Au deposition to the Boehmite surface is carried out through desired oblique angles. **c** SEM images showing alterations in quasi-parallel alignment of anisotropic Au nanostructures at deposition angle at 80°. Note that the angles at 45° or 0° cause no anisotropy of Au nanoparticles. The side and top views of the GNF substrates are shown in the left and right panels, respectively. Bar = 500 nm. Parallel-polar and Perpendicular-polar: the parallel and side directions of polarized light to generate SERS signals, respectively. White targets and arrows indicate the direction of the Au deposition. **d** A representative top-view SEM image with high magnification. Bar = 100 nm. **e** A representative picture depicting differences in SERS spectra of rhodamine 6 G (R6G) between GNF and GNC substrates. Comparison of the intensity at 1360 cm⁻¹ revealed that GNF substrates provides evidence that GNF-mediated SERS enhancement is 18.6-fold greater than that mediated by GNC

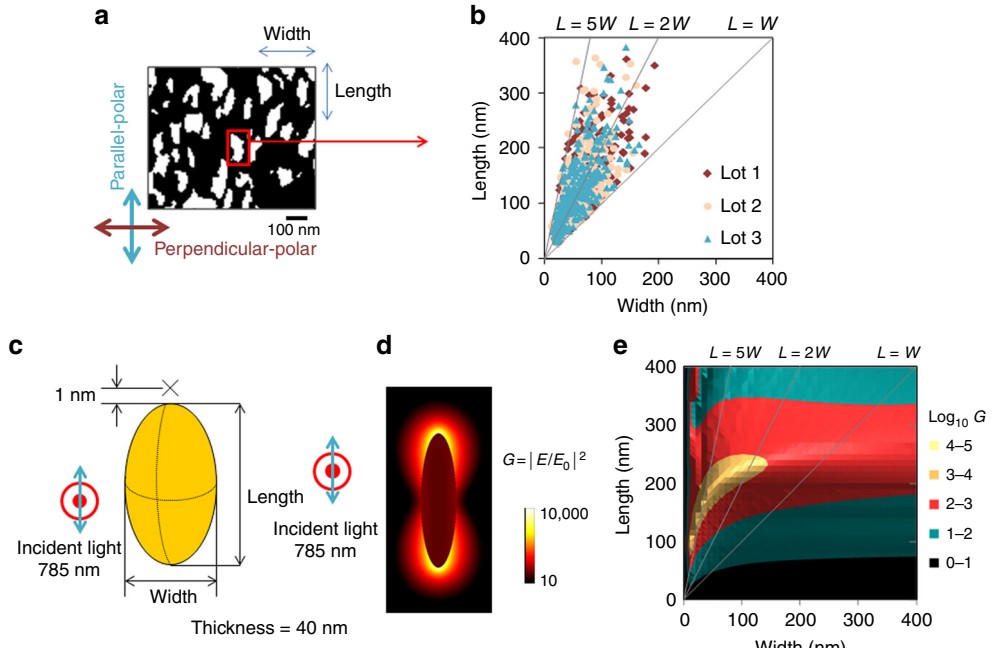

**Fig. 2** Finite-difference time-domain (FDTD) simulation for reasoning mechanisms by which electromagnetic fields are enhanced around the individual anisotropic Au nanostructures of GNF substrates. **a** A representative binary image collected from a top-view SEM image. **b** Distribution of lengths of individual Au nanoparticles (a red square in **a** indicates a representative one) as a function of their widths. Note that, in separate analyses using 3 different lots of the GNF substrate, a major population of the nanoparticles are plotted in a range between the lines $L = W$ and $L = 5 W$. **c** A simple model of the Au nanoparticle. Red point and circle indicate the direction of the incident light, while the blue arrow shows the direction of polarization of the light. The thickness of the Au nanoparticle was fixed at 40 nm, and the points at 1-nm far from the virtual surface of the particles were determined for simulating electromagnetic fields. **d** A representative simulation of electromagnetic field in and around a representative model of a single Au nanoparticle with the $L/W$ values of 3.8 ($L = 190$ nm, $W = 50$ nm) and the thickness of 40 nm, indicating the log $G$ value of 4.7. **e** Enhancement of electromagnetic fields and its relationship with $L/W$ ratios. Note that Au particles with the $L/W$ ratio in a range between 2.0 and 5.0 exhibited greatest enhancement of electromagnetic fields (yellow in Fig. 2e)

processes requiring no labelling. Furthermore, careful characterization of individual GNF-SERS signals enabled identification of glutathione (GS) at 298 cm$^{-1}$ or hypotaurine (HT) at 978 cm$^{-1}$, and retinoids at 1150 cm$^{-1}$. GS in T increased with development of cancer metastases, and was suppressed by knockdown of CD44 expressed in cancer cells, a molecule stabilizing cystine transporter (xCT) as part of cancer survival[11,12]. Interestingly, although biosynthesis of GS and HT requires cysteine as a substrate[13,14], HT, but not GS, in cancer cells serves as a robust CD44/xCT-independent anti-oxidant that is rapidly converted to taurine, and compensates for the decreased capacity of cancer to attenuate oxidative stress during CD44-knockdown-induced GS suppression.

## Results

### Characterizing GNF substrates for large-area SERS imaging.
We previously fabricated gold-nanocoral (GNC) substrates to perform large-area SERS imaging[15]. However, detection of larger numbers of metabolites with better sensitivity and selectivity was necessary. We therefore developed highly sensitive SERS substrates for Raman spectroscopy (Fig. 1a): the Au deposition from a perpendicular direction to the substrate (0° in the deposition angle in Fig. 1b) initiated growth of Au-nanospheres on the ledge of a boehmite structure (Fig. 1c: GNC substrate). Use of an oblique Au deposition technique[16] at an angle of 80°, but not of 45° or 0°, resulted in highly porous and anisotropic island growth of Au nanoparticles through their self-organized assembly without use of lithographic techniques (Fig. 1c, d). Scanning electron microscopic (SEM) images showed a random array of the GNF substrate, which is named after the anisotropic shape of the Au

nanostructure, as it resembles the shape of horse beans, and is distinct from spheroid GNC[15]. SEM (Fig. 1c, d) revealed that the interparticle space between Au nanoparticles formed by the deposition angle of 80° is larger than that of 0° and 45°. This morphological feature improved GNF-SERS sensitivity by increasing metabolite accessibility. Optimization of the GNF design (Supplementary Note 1 and Supplementary Fig. 1)[17] yielded a SERS enhancement by 18.6-fold versus GNC (Fig. 1e).

**Simulation of electromagnetic fields around a GNF unit.** To understand mechanisms by which GNF substrates enhance SERS signals, we attempted to determine the anisotropic features of the nanostructure of an individual GNF. Binary images of top-view SEM images (Fig. 2a) displayed heterogeneously elliptical regions as bright areas (a red square in Fig. 2a), and thus enabled to measure maximum lengths (parallel to the direction of Au deposition) and widths (perpendicular to the direction of Au deposition) of individual Au nanoparticles. These measurements in three separate lots of GNF substrates showed that length was greater than width in most GNF units and that the average length/width (L/W) ratio was 1.0:5.0 (Fig. 2b), and enabled to simplify the morphological feature of Au nanoparticles in GNF (Fig. 2c), which is suitable for finite-difference time-domain (FDTD) simulation[18,19] to determine effects of various lengths and widths of model particles on enhancement of electromagnetic fields around individual Au nanoparticles. Local field enhancement ($G = |E/E_0|^2$) was greatest in the two polar apexes of a single unit (Fig. 2d). Superimposition of the local field enhancement (G in Fig. 2d, e) on the distribution of lengths and widths (Fig. 2b) showed that most of individual Au nanoparticles in GNF

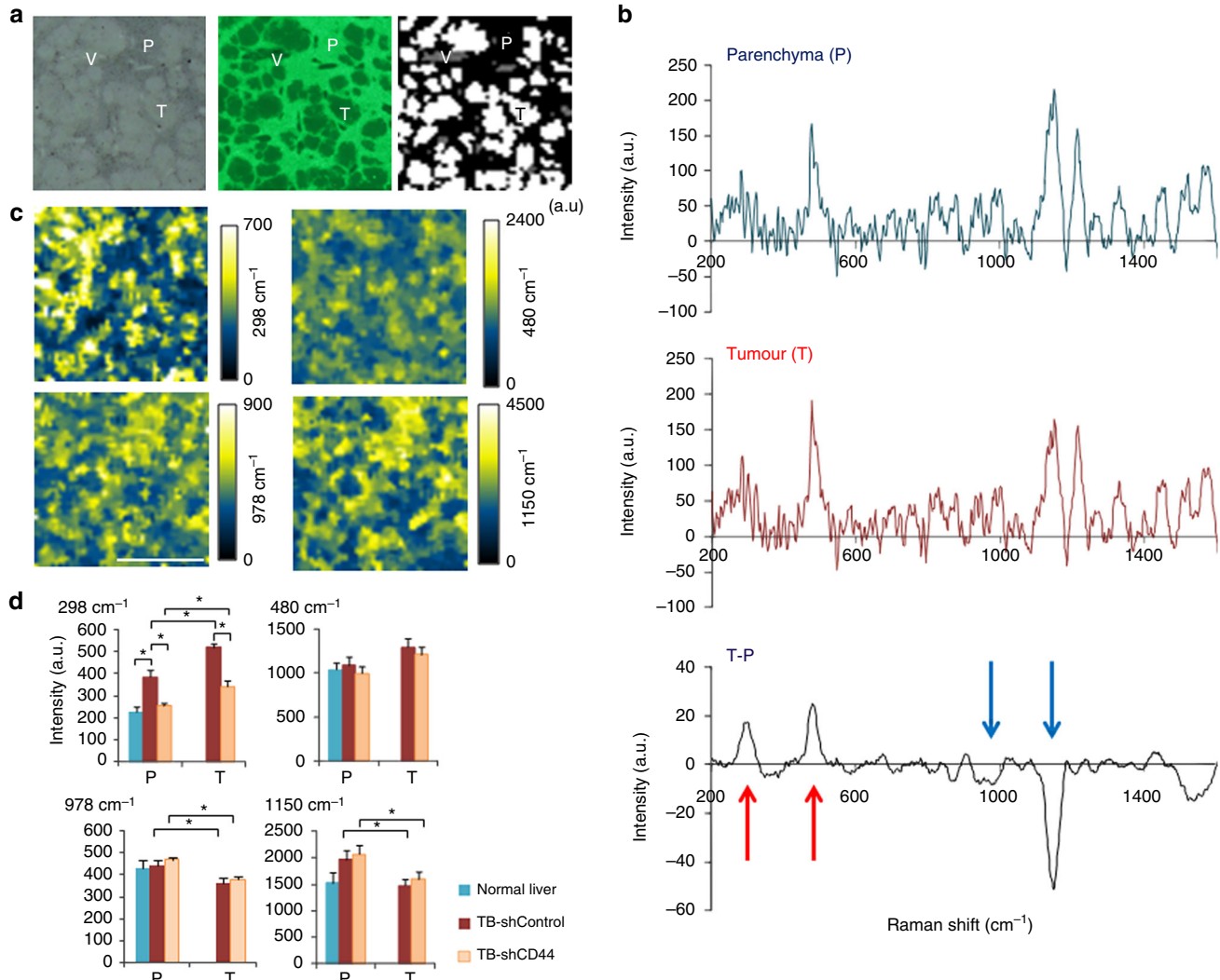

**Fig. 3** Screening of tumour- and parenchyma-specific SERS signals in HCT116 xenograft model in vivo. **a** A representative set of transmitted-light (gray scale) and auto-fluorescent images (green) of frozen tissue sections collected from a liver bearing metastases yielded by xenografts of human-derived HCT116 cancer cell lines. The right panel indicates a binary image to distinguish three different regions such as tumour (T: white), parenchyma (P: black) and vessels (V: gray). **b** A spectral profile of GNF-SERS signals in a range between $200\ cm^{-1}$ and $1600\ cm^{-1}$ in P- and T-regions, respectively. The data were averages of different tissue sections from tumour-bearing livers of 7 separate animals. Note that several peaks that are greater in tumours than in parenchyma (red arrows at $298\ cm^{-1}$ and $480\ cm^{-1}$), and those greater in parenchyma than in tumours (blue arrows at $978\ cm^{-1}$ and $1150\ cm^{-1}$). **c** Large-area SERS imaging of the 4 major peaks using GNF substrates. Signals were accumulated in a width of $20\ cm^{-1}$ at the median wave numbers of $298\ cm^{-1}$, $480\ cm^{-1}$, $978\ cm^{-1}$ and $1150\ cm^{-1}$. Intensities indicate arbitrary units (a.u.). Images were processed by a median filter. Bar = 1.0 mm. **d** Differences in intensities of the region-specific SERS signals in the xenograft-transplanted livers. The signal intensities were expressed as arbitrary units (a.u.). T and P indicate tumour- and parenchymal regions. Data indicated mean ± SE of 7 separate experiments. Statistical significance was analyzed by ANOVA with Fischer's LSD test. *$p < 0.05$ was considered statistically significant. $F$ values of 298, 480, 978, 1150 $cm^{-1}$ were $F_{(4, 28)}$ = 24.482, 3.039, 3.285 and 3.039, respectively

substrates were in regions indicating greater enhancement of electromagnetic fields (Fig. 2e). Among them, Au nanoparticles with the $L/W$ ratio between 2.0 and 5.0 and a length less than 250 nm exhibited the greatest enhancement of electromagnetic fields (Fig. 2e). Furthermore, Au nanoparticles with greater lengths and widths exhibited smaller $G$ values (green regions in Fig. 2e), suggesting that morphological anisotropy with optimal lengths of Au nanoparticles is crucial for improving SERS excitation sources. According to FDTD simulation, the local field enhancement by two adjacent Au nanoparticles of GNC are diminished by increasing the gap distance over 10 nm (typical gap distance of GNF in Fig. 2a), but the enhancement by those of GNF did not regress but stayed a plateau level (Supplementary Figs 2a-c). These results suggest that the anisotropy of Au nanoparticles

rather than effects of the gap of the Au-nanoparticle, which is also called junction, determine SERS enhancement in GNF substrates, being in good agreement with previous studies[20].

**Automated extraction of tumour boundaries**. As described in the Methods, three regions—T, P and vessels (V)—were manually extracted from binary images (Fig. 3a). Raman spectra between $200\ cm^{-1}$ and $1600\ cm^{-1}$ in regions P and T were subtracted to generate difference spectra between T and P (Fig. 3b). Two were higher in T than in P (red arrows at $298\ cm^{-1}$ and $480\ cm^{-1}$), and two were higher in P than in T (blue arrows at $978\ cm^{-1}$ and $1150\ cm^{-1}$). At these peaks, GNF-SERS imaging yielded actual images of the T-bearing liver (Fig. 3c). To generate these images, Raman shifts at specific pixels were accumulated and processed

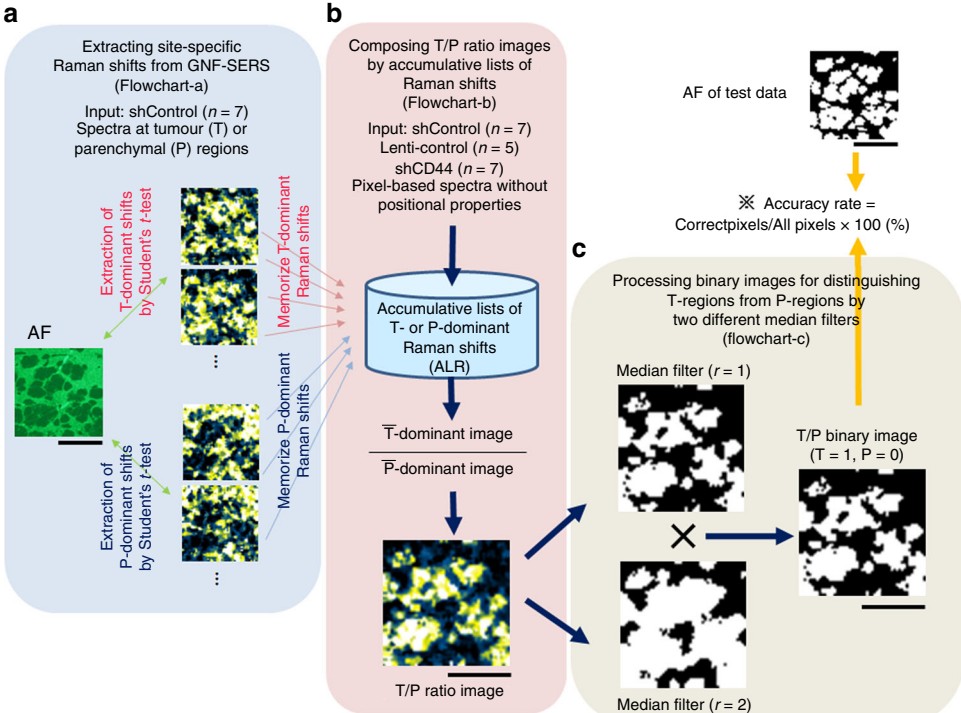

**Fig. 4** Tumour boundary extraction by differential SERS spectral analyses. **a** Selection of tumour (T)- and parenchyma (P)-dominant Raman shifts from shControl-xenografted liver tissues for building accumulative lists of T- and P-dominant Raman shifts (ALR) for automatic extraction of tumour boundaries. AF: autofluorescence. The detailed procedure is described in the flowchart of Supplementary Fig. 2a. **b** Establishment of ALR to compose the ratio images between T-dominant and P-dominant Raman shifts. T-bar and P-bar indicate averaged images of T- and P-dominant Raman shifts. The detailed procedures based on flowcharts are described in the flowchart of Supplementary Fig. 2a. **c** Schematic diagram of triangle auto-thresholding method to extract the boundaries between tumours and parenchyma. ($r = 1$: Median filter by 1 pixel; $r = 2$: Median filter by 2 pixels) Finally, the accuracy rate (%) was defined as the ratio of correct pixels versus total pixels ($51 \times 51$), and was calculated for individual images processed by flowcharts b and c. Bar = 1 mm

by a median filter (see Methods). Comparison of regions T and P indicated that raw SERS signals at $298\,cm^{-1}$ were significantly greater in T than in P. Raw signals in T and P were significantly attenuated by xenografting with HCT116 cells pretreated with shRNA against CD44, a surface antigen responsible for stabilizing xCT for cancer stemness and survival[11,12]. While the peak at $480\,cm^{-1}$ was marginally higher in T than in P (Fig. 3b, d) without statistical significance, the peak identification will be examined later. In contrast, the peaks at $978\,cm^{-1}$ and $1150\,cm^{-1}$ were significantly higher for P than for T (Fig. 3d). Interestingly, both peaks were insensitive to the CD44 knockdown in cancer cells. Three peaks displaying notable differences between regions T and P were further examined for subsequent identifications of specific metabolites later.

Cancer cells alter their metabolism to generate precursors of macromolecules necessary for their proliferation and survival. These alterations in metabolic systems may help distinguish T from surrounding tissues. We thus attempted to develop a GNF-SERS technique for identifying T boundaries on tissue sections. To that effect, we utilized all Raman shifts obtained from GNF-SERS. Unlike the protocols used for GNF-SERS imaging, Raman signal data were not accumulated, but were processed directly at the smallest detector resolution ($\sim2\,cm^{-1}$) for collecting many key Raman shifts to identify T boundaries (see Methods). The processed data showed 6 T-dominant shifts (T > P) and 22 P-dominant shifts (T < P) (Supplementary Table 1, flowchart-a in Fig. 4, and Supplementary Fig. 1a). Because GNF-SERS spectra involve numerous metabolite-derived Raman signals in T and P, statistical analyses of differences between T- and P-specific SERS spectra might benefit from the identification of T boundaries in tissue sections. We examined whether automated non-target

processing of such Raman signals aided in identifying spatial boundaries of T regions. The flowchart-a in Fig. 4 and Supplementary Fig. 3a was used to differentially accumulate T-dominant and P-dominant Raman shifts collected from liver tissues xenografted with shControl cells. This process resulted in building the accumulative lists of Raman shifts (ALR in flowchart-b of Fig. 4). Flowchart-b (Fig. 4 and Supplementary Fig. 3b) was used to extract T areas as very bright regions by calculating the ratio images between averaged T/P ratios (T-bar-dominant over P-bar-dominant images in flowchart-b of Fig. 4) in liver tissue xenografts in three groups (shControl, Lenti-control in which cancer cells were transfected with the virus vector alone, and shCD44; see Methods).

The image-processing procedure in flowchart-c (Fig. 4 and Supplementary Fig. 3c) was the final step for identifying boundaries. T/P ratio images collected from the three groups were processed with a newly developed method that uses 2 different median filters; one processes images through a radius of one pixel and yields sharply contrasted T boundaries with high noise levels. The other median filter processes images through a radius of 2 pixels and yields smooth and noiseless images with wider boundaries between T and P. These image sources were binarized to generate one-pixel- and two-pixel-based smoothened images (0 in P and 1 in T regions) and finally multiplied the two binarized images in which signals from T regions became 0 (flowchart in Supplementary Fig. 3c) (see Methods). The accuracy rates for the groups will be compared below. However, the present results suggest that image processing of global GNF-SERS improves automated identification of boundaries in T-bearing liver tissues.

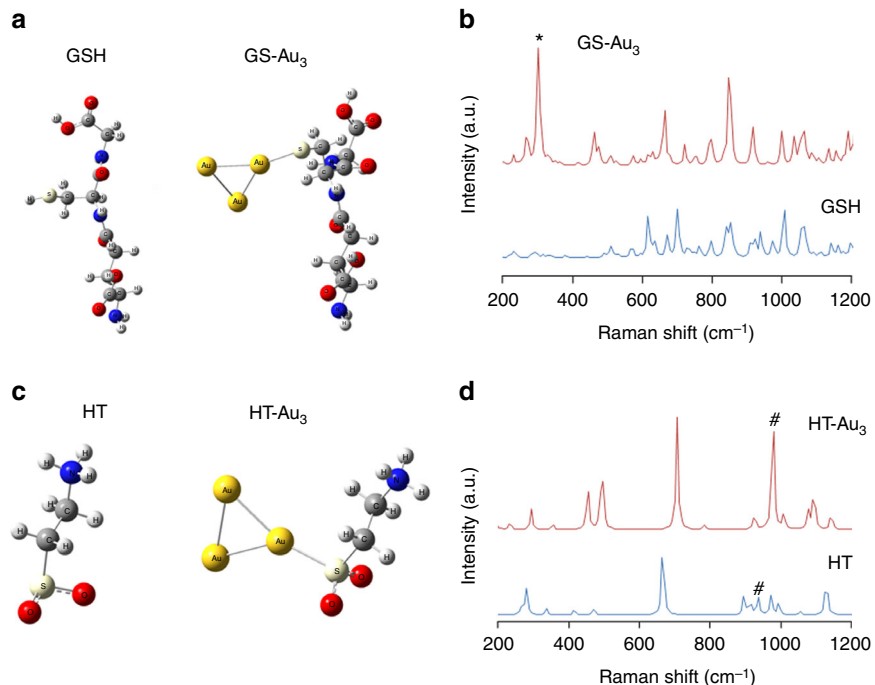

**Fig. 5** Non-empirical approaches to identify molecular vibration modes of reduced glutathione (GSH) and hypotaurine (HT) using quantum chemical calculation in silico. **a** Optimized geometry of GSH and $Au_3$-GS using B3LYP/6-311 + G*/SDD level, and **b** calculated Raman spectra. The Raman peak marked as * around 300 cm$^{-1}$ is Au-S stretching mode. **c** Optimized geometry of HT and $Au_3$-HT using B3LYP/6-311 + G*/SDD level, and **d** calculated Raman spectra. The Raman peaks marked as # around 970~980 cm$^{-1}$ are S = O stretching mode. Of note is that the peak in HT-$Au_3$ occurred at 977 cm$^{-1}$, being in good agreement with the peak for hypotaurine in solution as well as in SERS imaging (Fig. 3)

**Identification of GS and HT by GNF-SERS**. Identifying metabolites by SERS in vivo requires discrimination of many individual SERS signals yielded by various metabolites. The Raman shift at around 300 cm$^{-1}$ is responsible for Au-S stretching mode[21,22], and the present observation that the signal at 298 cm$^{-1}$ is sensitive to CD44 knockdown (Fig. 3d) led us to examine if the peak predominantly detected in T at 298 cm$^{-1}$ results from thiols. Thus, SERS signals for varied thiol metabolites were examined on GNF substrates after sufficient concentrations of analytes. These measurements utilized natural-dried aqueous solutions, and the relationship of the original concentrations of analytes to analyte densities on GNF substrates was described in Supplementary Fig. 4 and Supplementary Note 1. Physiologic concentrations of thiols other than GS, and taurine did neither generate the SERS peaks at 298 cm$^{-1}$ or at 978 cm$^{-1}$, as judged by the data collected by MS and imaging MS analyses in Supplementary Figs. 4-10 in Supplementary Note 1[23–30]. Furthermore, analyses of major liver metabolites did not generate Raman peaks at these wave numbers (Supplementary Fig. 11)[31]. These results suggest that GS accounts for SERS signals at 298 cm$^{-1}$. At the same time, HT is measurable by determining the SERS signal at 978 cm$^{-1}$.

Vacuum-MALDI imaging that enabled to visualize HT (Supplementary Fig. 10) suggested that HT serves as a putative metabolite occurring predominantly in P, and the result was consistent with topographic patterns of SERS imaging at 978 cm$^{-1}$ (Fig. 3c). These results raised a question to reason why T regions are enriched by taurine (Supplementary Fig. 8d) with apparently lesser amounts of HT; we speculated that cancer cells are armed with mechanisms that consume HT to be readily oxidized into taurine, leading to apparently low tissue amounts of HT.

**Quantum chemical calculation to characterize GSH and HT**. To determine whether the signals at 298 cm$^{-1}$ and 978 cm$^{-1}$ were derived from GS and HT, we used quantum chemical modeling to identify specific the molecular vibration modes of these metabolites in silico. This method allowed us to assign molecular vibration modes, which result from molecular interactions between GSH or HT and atomic Au surface (Supplementary Fig. 12 and Fig. 5 and Supplementary Note 1)[32–34]. First, to optimize the structure of a pseudo-Au surface for binding models for GS and HT, we simulated wave numbers of Au-S stretching mode generated by methyl-mercaptane with various numbers of Au-atom clusters ($CH_3S$-$Au_x$, Supplementary Fig. 12). The results indicating frequency peaks for Au-S stretching mode showed that the notable shift occurred between $CH_3S$-$Au_1$ and $CH_3S$-$Au_2$ and reached a plateau at 300 cm$^{-1}$ as the number of Au atoms increased (Supplementary Figs. 12b, c). A pseudo-Au surface for binding GS and HT was thus constructed by using triangular $Au_3$ structure in which Au–S interactions were stable.

The absence of Au did not generate notable GSH vibration modes around 300 cm$^{-1}$, but the presence of GS-$Au_3$ induced a strong Au-S stretching peak at 302.8 cm$^{-1}$ (Fig. 5a, b). The Au-free HT spectrum (blue line in Fig. 5d), generated a weak S = O stretching mode peak at 936 cm$^{-1}$, while the presence of $Au_3$-cluster enhanced and shifted the peak at 978 cm$^{-1}$ (Fig. 5c, d), which suggests that GNF-mediated SERS signals at 298 cm$^{-1}$ or 978 cm$^{-1}$ were produced by interactions of Au with GS or HT, respectively. We therefore assigned them to Au-S stretching mode (GSH) and S = O stretching mode (HT), respectively.

**GNF-SERS imaging of livers bearing colon cancer xenografts**. GS and HT are metabolites synthesized from cysteine provided from trans-sulfuration pathway and/or from xCT/CD44 that helps cystine incorporation from the extracellular space. CD44 interference by shRNA (shCD44) downregulates CD44s and its splicing variant CD44v in HCT116 cells, thereby suppressing T metastases[11,12,35] (Fig. 6a, b). Interestingly, shCD44 and

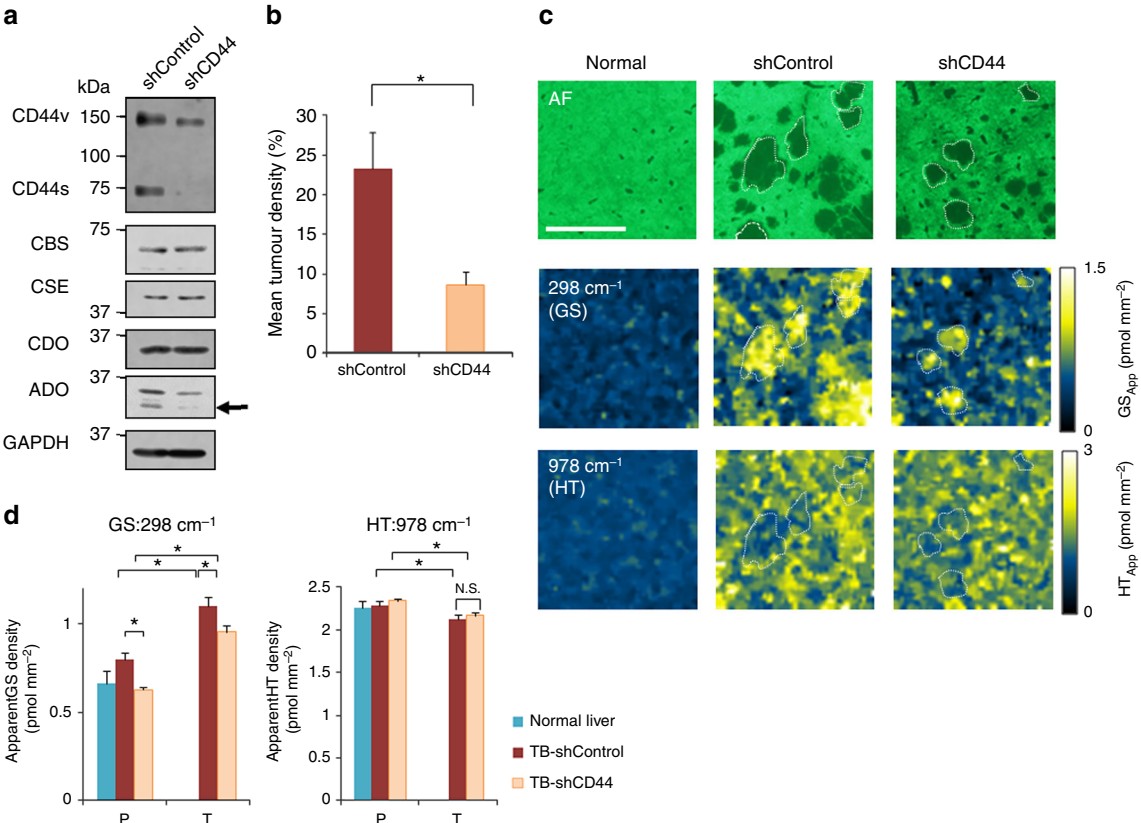

**Fig. 6** Semi-quantitative analyses of glutathione (GS) and hypotaurine (HT) in tumour-bearing livers. **a** Western blot analyses of CD44 expression in HCT 116 cells which were pretreated with control shRNA (shControl) or with CD44 shRNA (shCD44). CBS: cystathionine β-synthase. CSE: cystathionine γ-lyase. CDO: cysteine dioxygenase. ADO: 2-aminoethanethiol dioxygenase. Note that ADO is markedly down-regulated in shCD44 cells. GAPDH is an internal control. **b** Differences in % tumour regions in the tumour-bearing (TB) livers, showing that CD44 silencing causes tumour suppression. Statistical significance was analyzed by unpaired Student's *t*-test. *$P < 0.05$ versus the shControl group. $T(7) = 2.765$. **c** Representative pictures of large-area SERS images of glutathione (GS) and hypotaurine (HT) which were calibrated as apparent analyte density (AAD) for GS and HT, respectively. Bar = 1.0 mm. Dotted lines indicate location of several metastatic tumours. **d** Differences in apparent GS density and apparent HT density. P and T depict parenchyma and tumour regions, respectively. Blue: tumour-free controls, brown: shControl HCT116 xenografting and orange: shCD44 HCT116 xenografting. Statistical analyses were performed by ANOVA with Fischer's LSD test. *$Pp < 0.05$ was considered statistically significant. F values for comparing GS and HT were $F_{(4, 28)} = 18.287$ and 3.253, respectively

shControl cells expressed multiple trans-sulfuration enzymes including cystathionine β-synthase (CBS), cystathionine γ-lyase (CTH). An HT-synthesizing cysteine dioxygenase (CDO) was comparably expressed, while expression of another HT-synthesizing enzyme, 2-aminoethanethiol dioxygenase (ADO) was notably diminished along with decreases in ATP, energy charge and total GS in shCD44 as compared with shControl cells (Fig. 6a and Supplementary Table 2)[24,29,35–37]. Large-area GNF-SERS imaging achieved by the data calibration (Supplementary Fig. 4) indicated that GS and HT distributions were homogeneous among lobules in the normal group. In xenograft-transplanted livers, T exhibited significant increases in GS and decreases in HT as compared with the surrounding P (Fig. 6c, d). When transplanted with shCD44-treated HCT116 cells, GS signals in T were significantly suppressed. Interestingly, GS signals in P were also significantly decreased, which suggests that interference with CD44 in cancer suppresses GS in both T and P (Fig. 6c, d). In contrast, HT densities in P were unchanged in T-bearing livers xenografted with shCD44-HCT116 cells, and significantly greater than that in T (Fig. 6d).

Although significantly lower than in P, calibrated amounts of HT in T, as determined by GNF-SERS, was ~1.8 pmol mm$^{-2}$ in the apparent analyte density, which was comparable to the apparent analyte density of GS (Fig. 6d). This result is inconsistent with the findings of HT imaging by vacuum-type imaging MS, which showed that HT was undetectable in T (Supplementary Fig. 10). Furthermore, unlike GS, CD44-KD in the cancer cells did not cause a decrease in HT (Fig. 6). These results led us to examine the possibility of an alternative pathway for providing HT, one that does not depend on the xCT/CD44 complex.

**Cancer cells silencing CD44 require HT for survival.** We examined whether CE-MS data might underestimate accurate HT amounts in cancer cells, because of its artificial oxidation and/or its rapid turnover in T. Western blotting (Fig. 6a) showed that ADO was diminished; thus, shCD44 cells might utilize glucose to provide 3-phosphoglycerate (3-PG) and L-serine as a one carbon resource[38,39] for boosting HT synthesis and subsequent taurine synthesis through oxidative stress in the cells. We then used CE-MS metabolomics to identify differences between shControl and shCD44 cells in conversion of ¹³C₆-labelled glucose towards catalytic intermediates in glycolysis, serine metabolism, glycine cleavage pathway[38] and remethylation and trans-sulfuration pathways at different time points. Among metabolites, HT and taurine were determined with LC-MS analysis, which allowed us to accurately determine HT amounts by minimizing artificial oxidation of the metabolites (Fig. 7)[40,41].

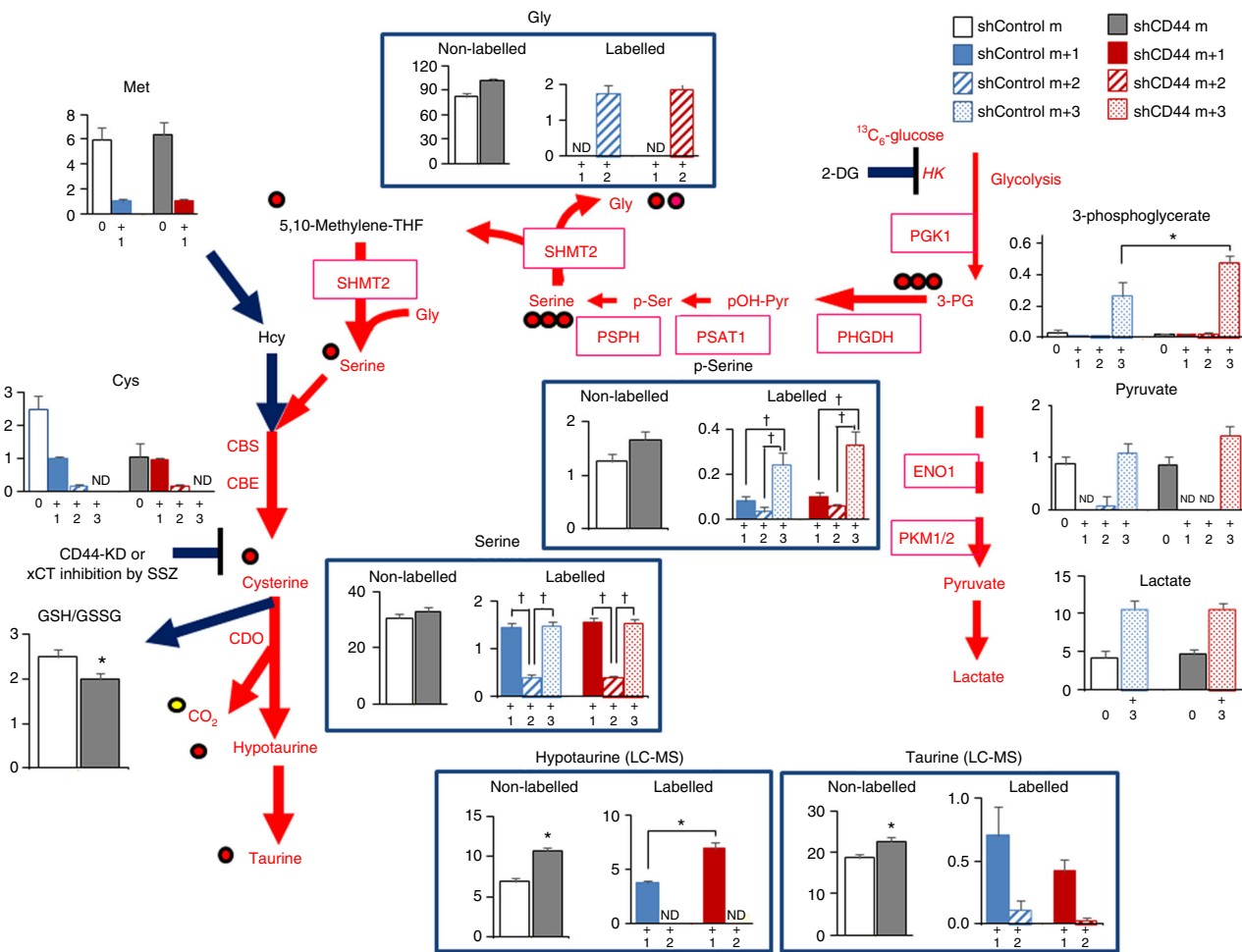

**Fig. 7** Differences in fates of $^{13}C_6$-glucose challenge between shControl and shCD44 cells in culture. The cells cultured in 1% FBS containing 5 mmol l$^{-1}$ $^{12}C_6$-glucose was removed and replaced with the same concentration of $^{13}C_6$-glucose for 15 min. Metabolites except for HT and taurine were determined by CE-MS (pmol mg$^{-1}$), while HT and taurine were determined by LC-MS to minimize oxidative modification of HT towards taurine. The units of HT and taurine contents were expressed as pmol mg$^{-1}$ protein. HK: hexokinase. 3-PG: 3-phosphoglycerate. PKM2: pyruvate kinase type-M2. PHGDH: phosphoglycerate dehydrogenase. PSAT1: phosphoserine aminotransferase 1, PSPH: phosphoserine phosphatase, SHMT: serine hydroxymethyl transferase. $^*p < 0.05$ analyzed by unpaired Student's $t$-test. $T$ values for statistical analyses are as follows: $T(14)$ for $^{13}C_3$-3-phosphoglycerate $= 3.537$, $T(14)$ for hypotaurine $= 7.354$, $T(14)$ for $^{13}C_1$-hypotaurine $= 5.622$, $T(14)$ for taurine $= 2.733$, and $T(14)$ for GSH/GSSG $= 2.169$. #$p < 0.05$ with ANOVA with Fischer's LSD test. $F$ values of p-Ser and Ser were $F_{(5,42)} = 6.519$ and $90.475$, respectively

After replacement with the culture medium containing 5 mmol l$^{-1}$ $^{13}C_6$-glucose for 15 min, shCD44 cells showed a greater elevation of $^{13}C_3$-3-phosphoglycerate (3PG) and $^{13}C_3$-phospho-serine, the metabolite that provides carbons into serine/glycine cleavage systems for shuttling a one methyl group via serine and cysteine towards HT (Fig. 7)[38,39]. In LC-MS analyses, cultured shCD44 cells significantly increased non-labelled HT and taurine at baseline, and $^{13}C_1$-labelled HT was detected as the early major product derived from $^{13}C_6$-glucose in shCD44 cells to a greater extent than in shControl cells. These results suggest that HT is a target supported by the one-methyl transfer system under CD44 knockdown. We also examined the effects of replacement with 5 mmol l$^{-1}$ $^{12}C_6$-glucose as a control for 15 min (Fig. 8a). The data showed significant increases in HT and taurine in shCD44 cells as compared with shControl cells. Furthermore, amounts of HT and taurine were suppressed by 2-deoxyglucose (2-DG). The acceleration of HT synthesis stopped at 60 min after replacement with $^{13}C_6$-glucose, and 2 different stable isotopic species of $^{13}C_1$- and $^{13}C_2$-taurine are increased (Fig. 8b), suggesting that both $^{13}C_1$- and $^{13}C_3$-serine serve as substrates for generating taurine. At 15 min, the relative amounts of

$^{13}C_1$- and $^{13}C_2$-labelled taurine versus non-labelled taurine were significantly lower. At 60 min the ratio was significantly reversed to show acceleration of HT oxidation to taurine during the later phase (Fig. 8c). Although the T HT signal in vivo (Fig. 6c, d) slightly increased in response to CD44 knockdown, in vivo and in vitro results were consistent: CD44 knockdown did not suppress amounts of HT in T or cultured cancer cells.

The accelerated HT/taurine synthesis in shCD44 cells coincided with significant induction of phosphoglycerate dehydrogenase (PHGDH) and phosphoserine aminotransferase-1 (PSAT1), and phosphoserine phosphatase (PSPH) and suppression of phosphorylated-pyruvate kinase M2 (p-PKM2 [S37], Fig. 8d–f). Although mechanisms remain unknown, altered expression of these enzymes by CD44-knockdown facilitates entry of glycolytic substrates into serine and glycine cleavage systems to compensate HT/taurine synthesis.

The early $^{13}C_1$-HT elevation followed by its disappearance and the increase in $^{13}C_1$- or $^{13}C_2$-labelled taurine led us to examine whether shCD44 cells survive when glucose utilization is blocked by 2-DG, a potent hexokinase (HK) inhibitor. As 2-DG increased, cell death was greater among shCD44 cells (Supplementary

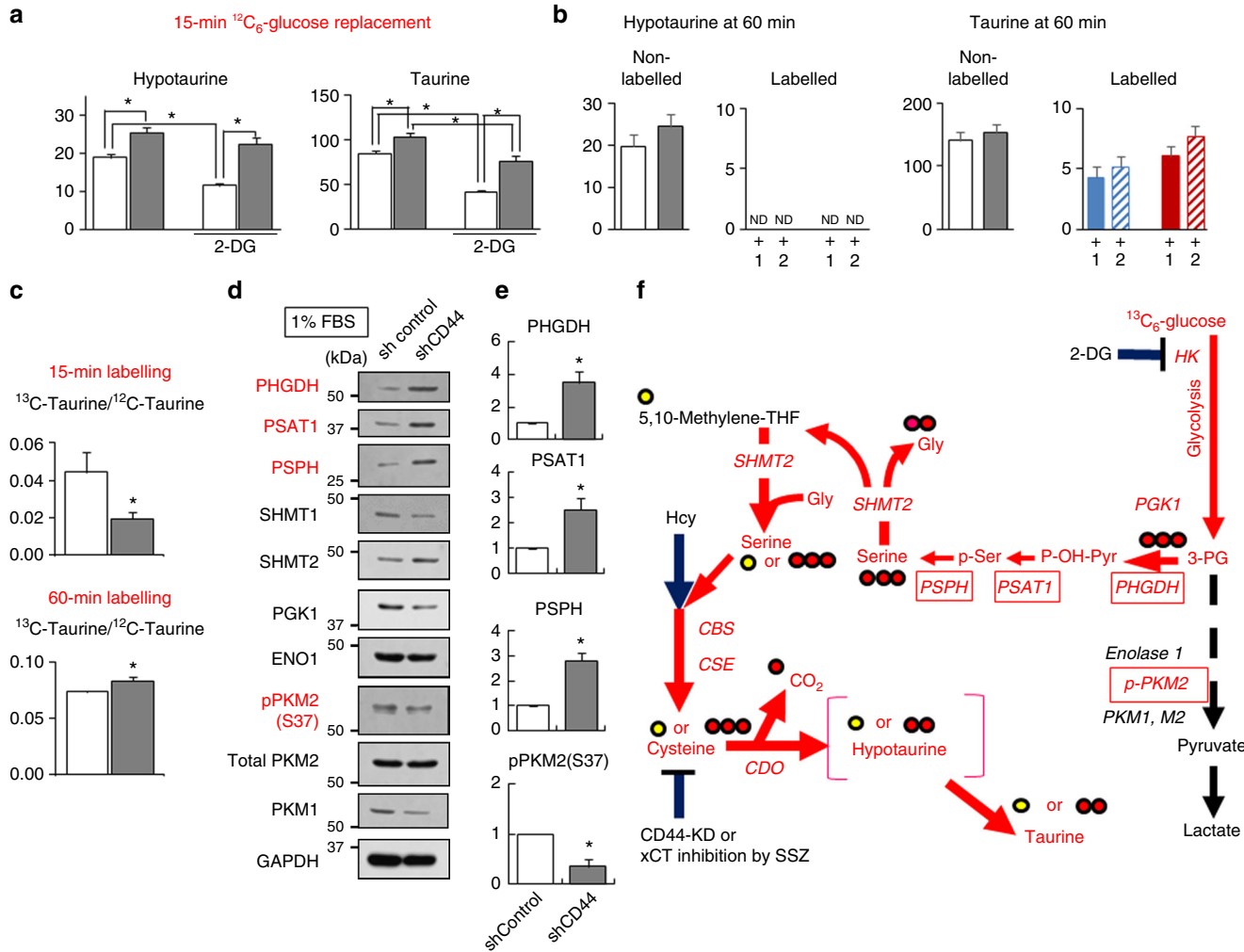

**Fig. 8** Altered expression of enzymes in glycolysis and serine/glycine cleavage system in shCD44 cells in culture. **a** Effects of replacement of the culture medium with 5 mmol l$^{-1}$ $^{12}C_6$-glucose on contents of HT and taurine as a negative control experiments of Fig. 8. Measurements were based on LC-MS analyses. All the units of HT and taurine contents were expressed as pmol mg$^{-1}$ protein. Note that 10 mmol l$^{-1}$ 2-deoxyglucose (2-DG) significantly suppressed amounts of HT and taurine in both shControl and shCD44 cells. *$p < 0.05$ with ANOVA with Fischer's LSD test. $F$ values of HT and taurine were $F_{(3,28)} = 21.273$ and 36.701, respectively. **b** Differences in fates of $^{13}C_6$-glucose challenge between shControl and shCD44 cells in culture at 60 min. The cells cultured in 1% FBS containing 5 mmol l$^{-1}$ $^{12}C_6$-glucose was removed and replaced with the same concentration of $^{13}C_6$-glucose for 60 min. Note that $^{13}C$-labelled HT disappeared at 60 min, while the $^{13}C_1$- and $^{13}C_2$-labelled taurine were elevated. **c** Comparison of $^{13}C$-labelled taurine versus $^{12}C$-taurine between those measured at 15 min and at 60 min. *$p < 0.05$ versus shControl (unpaired Student's $t$-test). **d** Altered expression of enzymes. Note that phospho-glycerate dehydrogenase (PHGDH), phosphoserine aminotransferase-1 (PSAT1) are increased, and pPKM2 (S37) are decreased in shCD44 cells as compared with those in shControl cells. **e** Relative values of densitometry in shCD44 data versus those in shControl data were analyzed by unpaired Student's $t$-test. Mean ± SE of 3 separate experiments: $T(4) = 3.595$ for PHGDH, and $T(4) = 3.370$ for PSAT1, $T(4) = 5.173$ for PSPH, and $T(4) = 4.060$ for pPKM2(S37). **f** Schematic diagram of metabolic systems in glycolysis, serine/glycine cleavage system, and trans-sulfuration pathway. 2-DG: 2-deoxy-glucose. SSZ: sulfasalazine, a potent xCT inhibitor

Fig. 13a). A fluoroprobe sensitive to reactive oxygen species (ROS) revealed that shCD44 cells at basal levels had significantly greater ROS than did shControl cells (Supplementary Fig. 13b). ROS levels were further enhanced by 2-DG treatment in shCD44 cells, and suppressed by exogenous HT application (Supplementary Fig. 13b). We also examined effects of sulfasalazine (SSZ), a potent xCT inhibitor[11], and found that shControl, but not shCD44 cells were sensitive to SSZ (Supplementary Fig. 13c). This result was consistent with the observation that CD44 knockdown destabilizes xCT expressed on cancer cell membranes[11,12,35]. Collectively with the data showing CDO expression in T and P (Supplementary Fig. 13d), these results suggested that shCD44 xenografts compensate for HT/taurine synthesis by inducing

PHGDH, PSAT1 and PSPH. Thus, HT is a robust CD44-independent anti-oxidant in cancer cells (Fig. 6c, d).

Many cancer cell lines expressing CD44 stop proliferation or lose viability by stable knockdown of CD44. Among 5 different cancer cell lines, MCF7 (CD44-low breast cancer) and MDA-MB-231 (CD44-high breast cancer) expressed either PHGDH or PSAT1, and PSPH. (Supplementary Figs. 14a, c). MCF7 and MDA-MB-231 are considered cell lines that showed comparable responsiveness of ROS to SSZ and 2-DG analogous to HCT116 cells (Supplementary Fig. 13). In these cell lines, 2-DG-induced ROS enhancement was attenuated by exogenously administered HT (Supplementary Fig. 14c), suggesting that HT cancels 2-DG-elicited ROS enhancement in these cells, as well.

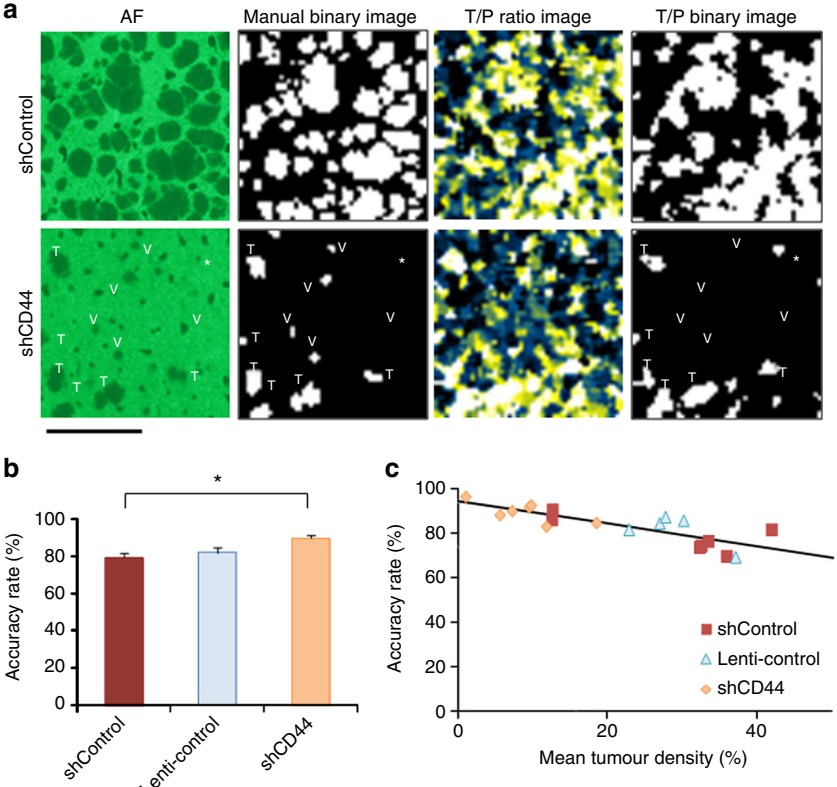

**Fig. 9** Robustness of automatic tumour boundary extraction in shCD44-xenografted livers. **a** Representative pictures indicating automatic image processing from manual binary images towards T/P binary image to calculate the accuracy rates of tumour boundary extraction in shControl- and shCD44-xenografted livers. Bar = 1 mm. The accuracy rates of the representative pictures in shControl and shCD44 images were 67.9% and 93.5%, respectively. T and V indicate location of tumours and vessels (e.g. central venules) in the representative microscopic field. An asterisk indicates a venule recognized in the autofluorescence (AF) image which turned out to display the presence of tumour after analyzing tumour boundaries. **b** Differences in accuracy rates among shControl, Lenti-control and shCD44 groups. Note that the accuracy rate in the shCD44 group is significantly higher than the shControl group. ANOVA with Fischer's LSD test. *$p < 0.05$ versus shControl. $F_{(2, 16)} = 4.970$. **c** Plotting accuracy rates of individual images as a function of tumour occupancy rates among different groups, showing a negative correlation ($Y = 0.5088\ X + 94.705$; $R^2 = 0.664$, $p = 0.00002$). Brown, light blue and yellow data points indicate those from shControl, Lenti-control and shCD44 groups, respectively

**Robustness of T boundary extraction in shCD44-xenograft.** Differences between shControl and shCD44 cells in CE-MS-based metabolomics (Supplementary Table 2), $^{13}C_6$-glucose-dependent flux analyses (Fig. 7) and GNF-SERS imaging (Fig. 6) led us to investigate whether the parent database of Raman shifts (Fig. 4 and Supplementary Table 1) collected from shControl cells was usable to identify T boundaries in shCD44-xenografted liver tissues in which metabolic profiles largely differ. Surprisingly, the accuracy rate of T boundary identification in shCD44-xenografted livers was significantly higher than those for shControl- and Lenti-control-xenografted livers. The accuracy rates of individual images as a function of T occupancy rates among the groups shows a significantly negative correlation (Fig. 9a, c). When the cross-sectional percentages of T density were less than 20% (<0.8 mm²), the accuracy rates exceeded 80%. Furthermore, this method of identifying T boundaries allowed us to differentiate large V (e.g. central venules shown by "v" in Fig. 9a) from metastatic T in the sections, because T, but not V, produce multiple molecular vibration modes for many metabolites, thus becoming detectable by GNF-SERS imaging. These results suggest that our method has benefits for automated detection of small metastases that are surrounded by sufficient parenchyma.

**GNF-SERS imaging in a mouse syngeneic glioblastoma model.** Profiles of metabolites differ among varied organs. It is worth examining whether our GNF-SERS imaging enables to extract differences in T-dominant and P-dominant metabolites in a mouse syngeneic model of glioblastoma derived from genetically engineered cancer stem cells, which displayed complicated pathologic features analogous to human glioblastoma such as inflammatory responses and gliosis[42–44]. We attempted to extract T borders by a professional pathologist using hematoxylin-eosin (HE)-stained tissues (Fig. 10). As seen, T regions (red in the right lower panel of Fig. 10a) seemed wider than binarized GFP-positive regions (the left lower panel of Fig. 10a) presumably because of the presence of the intermediate regions containing cancer cells with weak GFP expression that coincided with inflammatory responses and/or gliosis (Fig. 10a). The difference spectra of GNF-SERS signals between T regions determined by the pathologist and P regions in striatum (blue in the lower right panel in Fig. 10a) indicated 2 major peaks at 485 cm$^{-1}$ and 726 cm$^{-1}$ (arrows in Supplementary Fig. 15), which indicated localization of the GNF-SERS signals in T regions (Fig. 10b–d).

To extract T boundaries, we analyzed GNF-SERS spectral data without accumulation, and succeeded in extracting 16 T-dominant and 10 P-dominant shifts (Supplementary Table 3) to build ALR (Fig. 4) from 4 separate T-bearing mice. Using the flowcharts shown in the flowchart-b of Fig. 4, the data from the separate T-bearing brain slice were input to compose the T boundary extraction by the binary processing protocols

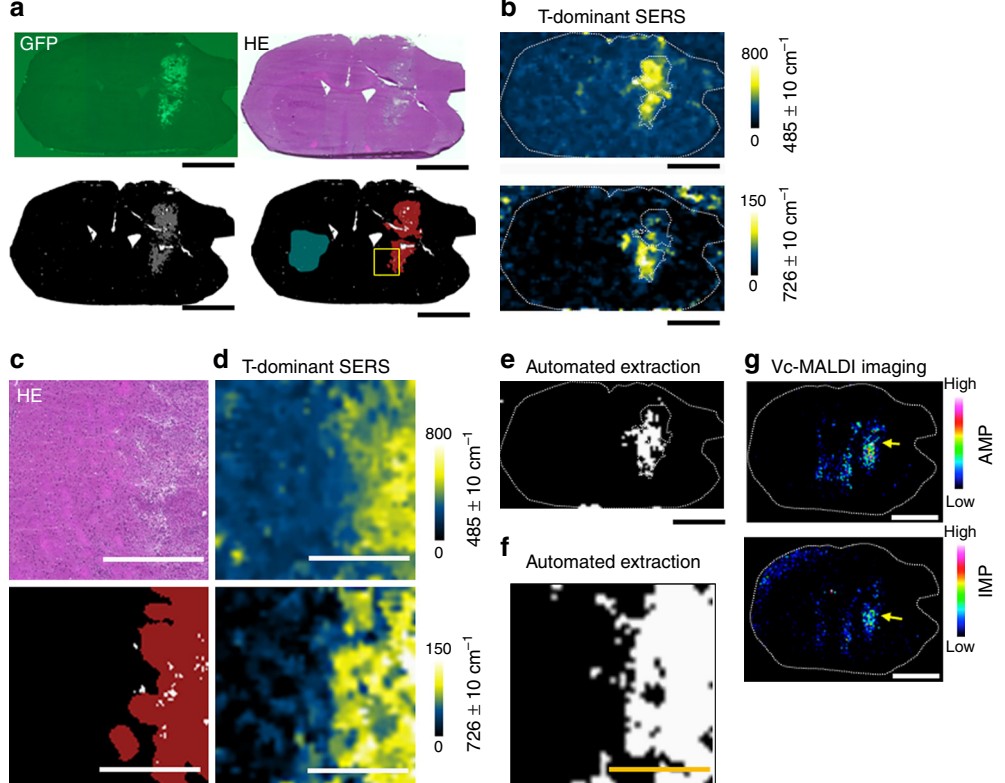

**Fig. 10** Tumour-specific metabolic profiles revealed by GNF-SERS in a syngeneic glioblastoma model. **a** Representative low-magnification images of glioblastoma-bearing brain. GFP: the fluorograph, HE: Haematoxylin-eosin staining. Lower panels show classification images of the upper images. Dark red and blue regions in the right lower panel indicated main tumour regions and the contralateral striatum region which were manually determined by a professional pathologist, respectively. Bar = 2.0 mm. Yellow squares show the regions indicating in **c**, **d**. **b** Representative pictures of GNF-SERS imaging showing T-dominant peaks at 485 cm$^{-1}$ and 726 cm$^{-1}$ in the same brain slices. Spatial resolution: 100 μm. Bar = 2.0 mm. **c** Magnified HE images and the main tumour region determined by a professional pathologist (highlighted in red). White spots: cracks and small defects in the tissue section. **d** Representative magnified images of T-dominant SERS peaks at 485 cm$^{-1}$ and 726 cm$^{-1}$ showing the boundary between glioblastoma and the surrounding tissue. Spatial resolution: 25 μm. Bar = 0.5 mm. **e** Automated extraction of tumour boundaries in the whole brain slice. Bar = 2.0 mm. **f** Automated extraction of tumour boundaries under high magnification. Bar = 0.5 mm. These binary images were T/P binary images (T = 1, P = 0) which were processed by the method shown in Fig. 4 and Supplementary Fig. 3. **g** Adenosine and inosine monophosphates (arrows) visualized by vacuum-type MALDI mass imaging (Vc-MALDI) in the serial frozen section. Bar = 2.0 mm

(flowchart-c in Fig. 4) enabled the boundary extraction of glioblastoma (Fig. 10e, f).

As previously reported[15], the peak at 726 cm$^{-1}$ seemed to be yielded by metabolites with adenine ring (e.g. AMP or IMP), as judged by vacuum-type MALDI imaging in the serial section showing enrichment of these metabolites in T-regions (Fig. 10g and Supplementary Fig. 15b). Previous studies[45] showed that S-S stretch of hydrogen polysulfides yields the peaks at around 480 cm$^{-1}$ by conventional Raman spectrometry. We thus examined whether the peak at 485 cm$^{-1}$ in the T-bearing brain and that at 480 cm$^{-1}$ in liver xenografts (Fig. 3b, d) derived from endogenous polysulfides[46,47]. Among small molecular hydrogen polysulfides ($H_2S_x$, $x = 1$–$4$) which are commercially available $Na_2S_x$ ($x \geqq 2$) generate the peaks at 280 cm$^{-1}$ (Au-S)[45] and 455~480 cm$^{-1}$ (a mixture of central and terminal vibration modes of S-S stretch[48]) (Supplementary Fig. 15c). SERS spectra yielded from $Na_2S_3$ and $Na_2S_4$ also showed the broad peaks at 280 cm$^{-1}$ and 460 cm$^{-1}$ (Supplementary Fig. 15d). Whether these polysulfides are detectable in T-containing brain tissues by LC-MS was examined by the monobromo-bimane-induced derivatization method (Supplementary Note 1 and Supplementary Methods)[37,49]. The data suggested that the glioblastoma-bearing ipsilateral hemispheres exhibited greater amounts of sulfide dibimane than the glioblastoma-free control hemispheres

in vivo (SDB in Supplementary Fig. 15e). Although discernible molecular entities of the endogenous polysulfides remain unknown, these data suggest that the GNF-SERS imaging serves as a useful device to visualize endogenous polysulfide compounds.

## Discussion

The current GNF-SERS technique provides excellent SERS excitation sources, thereby enabling large-area imaging and profiling of metabolic systems in T-bearing tissues. The robust, spatially uniform enhancement of GNF signals that covers extensive areas of tissues exceeds the performance characteristics of previously reported SERS substrates.[2] The present technique covers the mesoscopic areas between the macroscopic and cellular levels of bioimaging and improves our understanding of the metabolic interplays between micrometastases and host tissues. This method is a breakthrough in automated processes for collecting SERS spectroscopic data and identifies metabolic differences between T and their surrounding tissues. Mass spectrometry[50,51] and conventional Raman spectrometry[3,52] face challenges in identifying cancer in biopsy or intraoperative specimens. However, the current study first to suggest that global characterization and imaging of GNF-SERS signals, which contain numerous molecular vibration modes of metabolites, might be useful for

automated identification of T boundaries in tissues. Another breaking-through of highly sensitive GNF-SERS is the identification of several metabolites such as GS at 298 cm$^{-1}$, HT at 978 cm$^{-1}$ and retinol and/or retinol palmitate at 1150 cm$^{-1}$ as a biomarker signature for differentiating T regions from surrounding tissues in a mouse model of liver metastases of human-derived colon cancer xenografts. GNF-SERS imaging in the syngeneic mouse model of glioblastoma raised the possibility that polysulfides and metabolites with adenine ring constitute T-dominant compounds which yields the peaks at around 480 cm$^{-1}$ and 726 cm$^{-1}$, respectively. Although further investigation is necessary to determine the exact molecular entities of polysulfides in T-bearing livers[53] and brains[54], the current results suggest that GNF-SERS serves as a method to visualize endogenous polysulfides in vivo.

Light-sensitive metabolites[26,27,55], or those with ROS-sensitivity (e.g. GS, HT and polysulfides) are difficult to investigate, as they are labile biomarker molecules prone to artificial oxidation by technologies that use high-energy laser and electric fields (e.g. MALDI and CE-MS). Studies using MS technologies might therefore overlook the possibility that the metabolic flux to generate HT is present, but undetectable by both AP-type or vacuum-type MALDI-MS imaging because of its instability against laser sensitivity or rapid consumption of HT in cancer cells. Use of GNF-SERS under infra-red laser excitation prevents labile HT from oxidation into taurine, and thus helps detect HT in T regions, while HT signals in P are stronger. HCT116 cells express CDO (Fig. 6a and Supplementary Fig. 13d), a rate-limiting enzyme for HT biosynthesis, and execute glucose biotransformation into HT and taurine (Fig. 7). The mechanisms involve glycolysis and induction of PHGDH, PSAT1 and PSPH for serine/glycine-cleavage systems to provide one-methyl group for synthesizing HT, and its rapid conversion into taurine. This suggests that HT is a robust anti-oxidant consumed for cancer survival under GS-suppressing conditions. The greater oxidative stress in shCD44 cells than in shControl cells (Supplementary Fig. 13b) indicates that shCD44 cells rapidly incorporate $^{13}C_6$-glucose and convert it to $^{13}C_3$-3-PG and $^{13}C_3$-serine, and then to $^{13}C_1$-serine through glycine cleavage systems. Finally, as compared with shControl cells, a greater flux of $^{13}C_1$-HT is induced, which disappears rapidly through oxidation into taurine.

Among human-derived cancer cell lines tested in this study, MCF7 (CD44-low) and MDA-MB-231 (CD44-high) had similar ROS responsiveness to SSZ and 2-DG, and 2-DG-induced ROS elevation was dose-dependently inhibited by HT (Supplementary Fig. 14c); these results suggest that the protective effects of HT against cell death are shared by HCT116 cells, and support the hypothesis that upregulation of PHGDH, PSAT1 and/or PSPH is necessary to support HT-dependent protecting mechanisms (Supplementary Fig. 14a). The current study deserves further investigation provided for whole mechanisms by which HT protects cancer cells against oxidative stress through the actions of multiple enzymes such as PHGDH[38], PSAT1[56,57], PSPH and phosphorylated PKM2[39,58].

Finally, this study introduces the possibility of pathological identification of T boundaries—without labelling or staining—in two different cancer models. In the liver xenograft model, although CD44 knockdown attenuated the increase in GS, a key metabolite in differentiating T and P, CD44-silencing in cancer cells improved the accuracy rate of the boundary identification. Possible reasons for improved accuracy rates in the shCD44-xenografted group include the inverse values for P-dominant shifts yielded by biomarker metabolites (e.g. HT and retinols), other unidentified shifts in metabolites with differences between regions T and P, and the increased surface areas of P containing greater numbers of P-dominant Raman shifts (Supplementary

Table 1), which result from the relative shrinkage of T regions. Although the high spatial occupancy of T regions in the tissue areas of interests reduces the accuracy rates, our results suggest that use of GNF-SERS for automated comparison of metabolic profiling between T and P is beneficial for sensitive detection of relatively smaller cancer foci in tissues (Fig. 9). It should be noted that global metabolic reprogramming of colorectal cancer occurs at adenoma stage, and is induced by MYC[59].

In the mouse syngeneic glioblastoma model, which was characterized by complicated pathological pictures, the current algorithm for digital image processing (Fig. 4 and Supplementary Fig. 3) combined with pre-learning by an experienced pathologist might shed light on a possibility of automated pathological diagnosis of cancer. Detection of global metabolic profiles by GNF-SERS benefits understanding robustness of metabolic systems of cancer[60], at least in the two different models of cancer development in mice. However, under clinical settings, post-operative cancer specimens undergo an exposure to vascular ligation that might largely alter metabolomics of samples. Thus, further studies using clinical specimens are obviously necessary to determine whether SERS imaging is applicable to pathological diagnosis of cancer. Refinement of automated and non-labelled diagnostic GNF-SERS imaging for microscopic cancer detection is now underway in our laboratory.

## Methods

**Fabrication of the GNF substrates**. A $24 \times 24 \times 0.5$ mm$^3$ glass plate was washed with surface active agents (W304/LB93, ADEKA Corp.) to remove organic contaminants, and rinsed in ultrapure water using an ultrasonic bath. After drying the plate in spin drier, aluminum (Al) was deposited on the plate at 50-nm thickness using a reactive DC magnetron sputtering system (SPF-530H, ANELVA) at a deposition rate of 5 Å s$^{-1}$ (Supplementary Fig. 1a). The Al film was subsequently boiled for 15 min in hot water (100°C) to form boehmite (AlO(OH)), and was dried with N$_2$ gas according to our previous method[15] (Supplementary Fig. 1b). Au was deposited on the boehmite nanostructures by electron-beam evaporation system (EBX-8C, ULVAC) with different deposition angles of 0°, 45° and 80° (Supplementary Fig. 1c). As shown in Fig. 1b, c, we determined that 80° in the deposition angle and 110 nm in Au thickness, which was evaluated by using R6G and our custom-made inverted Raman microscopic system with ×20 (NA = 0.4) objective lens, gave optimal conditions such that SERS signal can be maximally enhanced. This microscope system was equipped with a fibre-coupled spectrograph (Ocean Optics, QE65000), and with an xy-scanning stage with highest resolution of 250 nm, and with a 785-nm diode laser as described previously[15]. The sensitivity and frequency of the microscope system were calibrated by the 520-cm$^{-1}$ Raman shift of silicon phonon mode before SERS measurements. The back-ground noise was subtracted from the recoded spectra by using the weighted mean fitting with Lanczos second function. All processes were conducted in a cleanroom. As described in the text, the coefficient of variation of the optimized GNF substrate in the uniformity was 11.0% ± 0.5%, and that in the lot-to-lot reproducibility was 2.1%.

**Morphological analyses of GNF substrates**. To determine periodicity of the boehmite nanostructure (Fig. 2a), binary maps were created from top-view SEM images. Sizes of the remaining particles were measured using the "analyze particles" function of ImageJ (Fig. 2). To evaluate lot-to-lot reproducibility of distribution and anisotropy of individual GNF units, measurements of lengths and widths of the Au nanoparticles were plotted as seen in Fig. 2b; results indicated that the L/W values of a major population of Au particles distributed in a range between 1.0 and 5.0 in 3 different lots.

**Numerical calculations of SERS enhancement factor using FDTD**. As described elsewhere[18,19], the local field enhancement (G) is represented by the ratio of the local field intensity ($|E|^2$) against the incident wave field intensity ($|E_0|^2$) at the wavelength ($\lambda$) of the excitation:

$$G = |E(\lambda)/E_0(\lambda)|^2$$

In the current study, a custom-made three-dimensional FDTD software was used to estimate these enhancement factors. Conditions of the simulation model to calculate G values were described as follows (Fig. 2c and d): the lengths and widths of the individual Au nanoparticles were variable in a range between 0 and 400 nm, and the thicknesses were 40 nm. A single Au nanoparticle was placed in air with a refractive index of 1.0. In the case of the local field enhancement, the particle was irradiated by the parallel-polarized plane-wave light at 785 nm. We calculated the

localized $G$ values at the apex position of the Au nanoparticle with a 1.0-nm distance from the Au surface, and the values were plotted as functions of lengths and widths of the particles (Fig. 2e). All these simulations were carried out with a grid size of 1 nm. The refractive index of Au at 785 nm was $0.177 + 4.54i$[61].

**Human-derived cancer xenografts in immunodeficient mice**. Animal experiments were carried out in accordance with the guidelines of Experimental Animal Committee of Keio University School of Medicine. Human colon cancer HCT116 cells were transfected stably with non-target control shRNA or with shCD44[11,12,35]. The cells were injected into the spleen of super-immunodeficient NOG (NOD/SCID/IL-2Rγ null) mice as described previously[12,29]. In brief, HCT116 cells were injected into the spleen of male NOG mice aged at 11–14 weeks at $1 \times 10^6$ cells/mouse. Two weeks after transplantation, liver lobes of the mice fasted for 17 h were excised under sevoflurane anesthesia and snap-frozen with liquid nitrogen. To examine T growth in vivo, 5-μm thickness cryo-sections were stained with hematoxylin and eosin (H&E). Age-matched control NOG mice without the xenograft transplantation were prepared for the control liver slices.

**GNF-SERS imaging of metabolites in vivo and in vitro**. To visualize metabolites in the frozen tissue sections, SERS imaging was carried out by using a GNF substrate, which is SERS-active substrate and above mentioned custom-made inverted Raman spectroscopic system[15]. To visualize metabolites on the tissues, cryo-sections with 5-μm thickness of the frozen livers were thaw-mounted on a GNF plate and kept in the vacuum dry chamber for 20 min. After capturing the images of the liver sections by an optical microscope (Nikon Eclipse E600: Nikon Corp., Tokyo, Japan) to select 2 mm x 2 mm square regions as regions of interests, the plate was set on the Raman microscope system. P were characterized by regions with positive autofluorescence, while T and large V are those indicating no autofluorescence. Since large V (e.g. central venules) were identified by determining cellular components (e.g. blood cells) in magnified transparent images, three different regions such as T, P and V were manually extracted to form a binary image (Fig. 3a) according to our previous method[62]. To capture autofluorescence images, a narrow-band filter (Olympus U-FGFP, excitation filter 460-480 nm, barrier filter 495–540 nm) was used to observe hepatocyte green autofluorescence[62]. Digital images were photographed with an Olympus DP73 camera. These sources of images were used to execute automatic T boundaries, as well as to identify T-specific and P-specific regions for visualzing and quantifying GNF-SERS signals that were shown in Fig. 3c, d and Fig. 6c, d.

For SERS imaging, we measured 2 mm x 2 mm square areas on liver tissues as regions of interests by using ×60 (NA = 0.7) objective lends. To compose two-dimensional SERS maps, spectra with the range of Raman shifts between 200 and 1600 $cm^{-1}$ were measured at $51 \times 51$ spots over a tissue covering the region of interests with a 40-μm-translation step. Acquisition time was 10 second for single spot. The diameter of beam spot was ~2 μm with ~1.0 mW power at a 785-nm laser. To achieve a quality of SERS images with greater signal-noise ratios, we accumulated SERS signals around the central peak wave number (typically ± 10 cm$^{-1}$). In case of HT imaging at 978 $cm^{-1}$, a width of the peak wave number was ± 8 $cm^{-1}$ (Supplementary Fig. 4b). This modification helped minimize the overlap with retinal and retinol palmitate. After that, SERS images were processed by a median filter ($r = 1$) to remove salt-and-pepper noise.

To determine SERS signals of the major metabolites in the mouse liver in vitro was carried out by the identical microscopic system used for visualizing tissue-derived SERS signals in vivo. In these experiments, solutions of varied concentrations of analytes were placed on a specially-arranged ring with 2-mm in diameter on the GNF substrate to form a 0.5-μl liquid droplet with a diameter of 2 mm, and natural-dried (Supplementary Fig. 4). The relationship between concentrations of the individual metabolites and the corresponding analyte densites was shown in Supplementary Figs. 4 and 11. When we examine a SERS spectrum of bovine serum albumin (BSA), we prepared 100 μl of the solution at a concentration of 1.008 g l$^{-1}$, which were coated over the entire GNF substrate with 173.6-μm thickness. After that the substrate was kept in the vacuum chamber for 20 min. In the middle of the drying process, the concentration of the BSA solution was 35 g l$^{-1}$ with 5 μm-thickness.

The experiments in vitro aimed to identify major SERS peaks of the whole Raman spectra of each metabolite in a range between 200 $cm^{-1}$ and 1600 $cm^{-1}$, and to establish calibration curves of SERS imaging signals for GS at 296 $cm^{-1}$ and HT at 978 $cm^{-1}$ (Supplementary Fig. 4). The curve fitting for calibrating glutathione or HT was carried out by establishing quadratic regression curves as shown in Supplementary Figs. 5a and c, respectively. When we used rhodamine 6 G (R6G) was used as an analyte (Supplementary Fig. 1), GNF substrates were immersed into an R6G solution (10 μmol l$^{-1}$) for 15 min, rinsed lightly by pure water, and dried in air according to our previous method[15].

**Establishment of control and CD44-knockdown HCT116 cells**. In this study, three different clones of human colon cancer-derived HCT116 cells were prepared; using expression vectors encoding a shRNA specific for human CD44 mRNA was obtained from Origene Technologies (Rockville, MD), we introduced it into HCT 116 cells by transfection with Lipofectamine 2000[11]. The sequence of shRNA used for CD44 knockdown (shCD44) throughout all studies was

5′- GCTGACCTCTGCAAG GCTTTCAATAGCAC-3′. The control non-targeting shRNA sequence was 5′-GCACTACCAGAGCTAACTCAGATAGTACT-3′, designated as shControl, unless otherwise mentioned, according to our previous reports[11,12]. The shRNA was obtained from Origene Technologies (Rockville, MD); we introduced it into HCT 116 cells by transfection with Lipofectamine 2000. The third clone of HCT116 cells was generated by transfection of Lentiviral vector control plasmids which were packaged by co-transfection with packaging plasmids in HEK293T cells by using BLOCK-iTR Lentiviral RNAi Expression System (Invitrogen) as described elsewhere[63,64]. Briefly, after 48-hr incubation, the Lentiviral particles were recovered from the medium for HEK293T cells, and were concentrated by ultracentrifugation. The HCT116 clone transfected with the control Lentiviral vector was designated as Lenti-control, unless mentioned otherwise.

**Tumour boundary extraction by GNF-SERS spectral analyses**. GNF-SERS spectra include SERS signals yielded by large quantities of metabolites in T and the surrounding P. Considering that metabolic systems in cancer cells are thought to be optimized for synthesizing macromolecules necessary for their survival, analyses of statistical differences between SERS spectra in T and those in P allowed us to generate lists of the differences in metabolite-derived SERS signals in the two different compartments in tissue sections. We thus hypothesized that accumulation of such machine-readable metabolite-derived Raman signals benefits to extract spatial boundaries of T regions automatically without any manual procedures in other tissue sections in which the same source of cancer xenografts is metastasized. To prove this hypothesis, we prepared two different clones of human colon cancer HCT116 cells. The first group of mice were transplanted with shControl in the spleen of NOG mice ($n = 7$). Their regional GNF-SERS spectral datasets were tagged with positional data of T or P which were determined by manual binarization.

To extract the boundaries between T and P, the data of Raman peaks were not accumulated, but processed directly at the smallest detector resolution (~2 cm$^{-1}$). The protocol of this kind benefits precise extraction of the specific Raman shifts (Supplementary Table 1) to distinguish T-dominant signals from P-dominant ones. This process was necessary to improve the accuracy rates of T boundary extraction as described later in this section. Accordingly, we established an accumulative list of T-dominant and P-dominant GNF-SERS shifts (ALR in Fig. 4) by using the data collected from the shControl group.

On the other hand, liver tissues metastasized with the xenografts derived from the second HCT116 clone named Lenti-control ($n = 5$) were used to examine whether lists of T-enriched and P-enriched metabolic fingerprints that were collected from the shControl group is suitable to execute machine-readable automatic extraction of the T boundaries in the frozen liver sections. To this end, we prepared two groups of xenograft transplantation model using different clones of human colon cancer HCT116 cells: The first was shControl ($n = 7$), which were used to establish the accumulative lists of the Raman shifts (ALR in Fig. 4 and flowchart-b of Supplementary Fig. 3a), in which data of Raman spectra were tagged with the positional data of T or P, which were determined by manual binarization. The second HCT116 clone named Lenti-control ($n = 5$) were used to examine whether the reference data collected from shControl cells enable to extract precise boundaries between T and P in tissue sections of interests which were collected from mice xenografted with Lenti-control. To that effect, spectral data of Raman shifts in the Lenti-control group were automatically sorted into T- and P-dominant shifts based on ALR (Supplementary Fig. 3b). All T- and P-dominant Raman shifts were averaged separately to build T/P ratio images that featured T regions in tissues. Finally, the T/P ratio images collected from the Lenti-control group were smoothened by using 2 different radiuses of median filters; one was to process images through a radius of 1 pixel, resulting in sharply contrasted T boundaries with high noise levels. On the other hand, another was to process them through a radius of 2 pixels, in which smooth and noiseless images were obtained, but with broadened boundaries between T and P. Using these image sources, we binarized these 1 pixel- and 2 pixel-based smoothened images (0 in P and 1 in T regions) according to the previously reported study using triangle auto-thresholding method[65], and multiplied the two binarized images in which signals out of T regions became 0 (flowchart in Supplementary Fig. 3c). We also analyzed GNF-SERS images of shControl- ($n = 7$), Lenti-control- ($n = 5$) and shCD44-xenografted liver tissues ($n = 7$) according to the flowchart (Supplementary Figs. 3b and c) to examine whether T derived from shCD44 cells metastasized in livers that are expected to change their metabolic profiles can be detected by using ALR database established in Supplementary Fig. 3a.

**MALDI imaging mass spectrometry (MALDI-IMS)**. In separate sets of experiments, we conducted atmospheric pressure or vacuum-type MALDI imaging according to our previous methods[12,29,66]. Details of methods were described in the text of Supplementary Note 1.

**Quantum chemical calculation to identify molecular vibration**. To identify molecules responsible for generating SERS signals with specific wave numbers, quantum chemical calculation was carried out according to previous studies[32–34]. The calculation was performed by Gaussian 09 Program package. The Raman

intensities were calculated using density functional theory (DFT) by using basis sets of functions of CPCM-B3LYP/6-311+ G*/SDD(Au)[32–34]. The vibrational frequencies were not scaled. All molecular geometries were fully optimized under assumption that the molecules are surrounded by $H_2O$ molecules.

**Cancer cell lines in culture and immunohistochemistry.** Besides HCT116 human colon cancer cell lines, we conducted in vitro experiments using several different cancer cell lines derived from human such as MCF7 that displays little CD44 expression, and MDA-MB-231, displaying CD44 (breast cancer cells), and PC9, H1650 and H2170, non-small cell lung cancer (NSCLC) cell lines expressing CD44v and resistant to sulfasalazine, an inihibitor of xCT (Supplementary Figs. 13a and b). The NSCLC cell lines, PC9, H1650 and H2170 were used in this study. PC9 cells were kindly provided by Dr. Fumiaki Koizumi (Tokyo Metropolitan Komagome Hospital, Tokyo). MDA-MB-231, H1650 and H2170 were purchased from the ATCC. MCF7 and HCT116 were purchased from RIKEN Cell Bank. Western blot analyses were conducted to examine expression of enzymes responsible for controlling glycolysis, serine/glycine cleavage systems and trans-sulfuration pathway. Details of the blotting methods and the methods to determine cell viability and oxidative stress were described in the text of Supplementary Note 1. Uncropped blotting data were shown in Supplementary Fig. 16. All cell lines were authenticated by STR profiling (Takara Bio, Otsu, Japan). Mycoplasma contaminations were tested in the cell lines using commercially available kit (MycoAlert™ Mycoplasma Detection Kit, Lonza).

**Chasing fates of $^{13}C_6$-glucose in cultured HCT116 cells.** We conducted CE-MS and LC-MS analyses to determine differences of $^{13}C$-labelled metabolites that belong to glycolysis, glycine cleavage pathway and remethylation and transsulfuration pathways between shControl and shCD44 cells in culture. In response to replacement with the culture medium containing 5 mmol l$^{-1}$ $^{13}C_6$-glucose for 15 min and for 60 min, $^{13}C$-labelled metabolites were generated in both cells (Fig. 8). At 15 min after exposure to $^{13}C_6$-glucose in the medium, $^{13}C_3$-phosphoserine derived from $^{13}C_3$-3PG provided $^{13}C_2$-glycine and $^{13}C_1$ transfer towards tetrahydroforate (THF). On the other hand, $^{13}C_1$ of glycine is cleaved and transferred to generate 5, 10-methylene THF. As a result, a series of these reactions generates $^{13}C_1$-methionine and $^{13}C_1$-serine through the robust catalytic activity of serine hydroxymethyl transferases (SHMTs) (Figs. 7 and 8). In these experiments, HT and taurine were determined by LC-MS (LCMS-8030plus triple quadrupole mass spectrometer, Shimadzu, Kyoto Japan)[12,67,68].

**Orthotopic implantation of neural stem cells to mimic glioblastoma.** All experiments described below were performed in accordance with the animal care guidelines of Keio University. Neural Stem/Progenitor Cell Culture was established as described previously[42,43]. Briefly, 6-week-old male Ink4a/Arf null C57BL/6 mice (B6.129-$Cdkn2a^{tm1Rdp}$; National Cancer Institute, Frederick, MD) were euthanized with a lethal dose of pentobarbital to extract brains and to isolate samples from their subventricular zone (SVZ). The SVZ fragments were microdissected, washed, trypsinized and then mechanically dissociated. Primary NSCs/NPCs were maintained as sphere culture in Dulbecco modified Eagle medium (DMEM)/F12 (Sigma, St Louis, MO) supplemented with 20 ng ml$^{-1}$ epidermal growth factor (EGF; PeproTech, Rocky Hill, NJ), 20 ng ml$^{-1}$ basic fibroblast growth factor (PeproTech), B27 supplement without vitamin A (Invitrogen, Carlsbad, CA), 200 ng ml$^{-1}$ heparan sulfate, with penicillin and streptomycin at 37°C in 5% $CO_2$/95% humidified air. The retroviral vector construct was prepared as follows. Human H-RasV12 cDNA [44] (kindly provided by P. P. Pandolfi) was cloned into the retroviral vector pMXs-IG (kindly provided by T. Kitamura) and the resulting construct transfected into Plat-E packaging cells[42] using FugeneHD (Roche Diagnostics, Mannheim, Germany). Medium was replaced once after 24 h. Viral supernatants were collected and filtered with 0.45-μm cellulose acetate filters (Iwaki, Kyoto, Japan) 48 h after transfection. Supernatants were centrifuged at 12,000 rpm for 6 h at 4°C, and the viral pellet was resuspended in small volumes of NSC culture medium. To establish brain T–initiating cells, primary Ink4a/Arf null NSC/NPCs were infected with retroviral supernatants. The resulting GFP-positive cells, termed RasG-NSCs hereafter, were sorted by flow cytometry in NSC culture medium and used for implantation within 20 passages after establishment.

To carry out orthotopic transplants, female C57BL/6 mice aged 6 to 8 weeks were anesthetized and placed into a stereotactic apparatus assisted by a z-axis (David Kopf Instruments, Tujunga, CA). A small hole was bored into the skull 2.0-mm lateral to the bregma using a dental drill. One hundred thousand viable cells in 2 μl of Hank's balanced salt solution was injected into the right hemisphere 3 mm below the surface of the brain using a 10-μl Hamilton syringe with a customized 26-gauge needle. The injection was performed over 2 min, with an additional 2-minute pause before removing the needle. The hole was sealed with bone wax, and the scalp was closed using an autoclip applicator[42]. None of the animals developed neurological deficits during the follow-up period. At 15 days after the transplantation, brains were extracted to prepare frozen tissue slices that include both T and the surrounding brain tissues.

Serial tissue sections were prepared from the frozen blocks of T-bearing brains collected from individual mice. Using the coronal tissue slices, both SERS imaging and hematoxylin-eosin staining were carried out. All images were digitally stored in

an open software (Nanozoomer Digital Pathology, Hamamatsu Photonics, Hamamatsu City). We examined whether our GNF-SERS imaging enables to extract differences in profiles of T-dominant and P-dominant metabolites in a mouse syngeneic model of glioblastoma derived from genetically engineered cancer stem cells[42–44]. Since this glioblastoma model displayed complicated pathological features including gliosis and inflammatory responses in and around main T, we extracted T borders using manual extraction by a professional pathologist, who examined haematoxylin-eosin (HE)-staining images (T-region; the red regions in the right lower panel of Fig. 10a, c) in tissue sections through high magnification images provided by Nanozoomer digital pathology software. In these analyses, the regions containing accumulated inflammatory cells were excluded from T-regions. Since the contralateral regions displayed no metastases or invasion in this model protocol, we chose the contralateral striatum regions as the control P (P-regions; the dark blue region in the left hemisphere in the right lower panel of Fig. 10a). Control regions in P of the contralateral striatum (P-region) of these brain slices were defined as T-free regions in the contralateral hemisphere. Selected regions (T and P) were superimposed on SERS imaging data, which were visualized according to the procedures used for the xenograft-transplanted model of HCT116 cells. In these experiments, GNF-SERS images were collected under two different spatial resolutions; low-magnification images with a spatial resolution of 100 μm that covered the whole coronal plane of brain slices (Fig. 10b), and high-magnification images with a spatial resolution of 25 μm that covered T boundaries that often coincided with gliosis and/or inflammatory responses (HE staining in Fig. 10c). Separately, we determined tissue contents of $H_2S$/polysulfides to compare them between T-free and T-bearing brain tissues by using the monobromo-bimane-derivatization method[37,49,54].

Separately, we examined whether recombinant GFP protein yield GNF-SERS signals. The drying protocols were conducted to determine whether varied concentrations of recombinant GFP (abcam, ab84191) dissolved in 0.1 mol l$^{-1}$ phosphate buffer at pH 7.4 yielded GNF-SERS signals in vitro. To determine approximate concentrations of GFP versus protein contents, an anesthetized mouse bearing glioblastoma was treated with the in-situ freezing procedure, and the T-enriched region was sampled by cutting with a surgical knife. The fluorescence intensities of recombinant GFP solutions were measured with SpectraMAX M2 microplate reader (excitation/emission = 488 nm/530 nm). To determine the approximate amount of GFP in T tissues, we sacrificed a T-bearing mouse under anesthesia, and the amount of protein versus T tissue weight was 0.18 mg mg$^{-1}$ T tissue. As GFP turned out to constitute ~0.1% of the total protein amounts, we prepared recombinant GFP solution at a concentration of 0.18 μg ml$^{-1}$, and an aliquot of 100 μl was placed on the GNF substrate to be dried up. Under these conditions, the density of GFP became sufficient enough to surpass that of GFP in vivo. GNF-SERS signals yielded from the region of $24 \times 24$ mm$^2$ were determined by the current Raman spectroscopy. As seen in the bottom of Supplementary Fig. 4a, GFP did not yield any notable GNF-SERS signals.

**Statistical analyses.** All quantitative results are presented as the mean ± SE of independent experiments. The sample sizes of each experiment were determined by considering the institutional guidelines to minimize unnecessarily killed animals. In the mouse glioblastoma model, we examined whether all mice used for experiments displayed T formation in situ: there were no mice which did not display T formation with no excluded animals from the group. Neither randomization nor blinding procedure was performed in the current animal study. Statistical differences between multiple groups were first analyzed by analysis of variance (ANOVA) with Fisher's least significant difference. Differences between two groups were analyzed by unpaired Student's t-test. $P < 0.05$ was considered statistically significant.

**Data availability.** The relevant data that support the findings of this study are available from the corresponding author upon reasonable request.

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

## Acknowledgements

The work was partly supported by JST, ERATO, Suematsu Gas Biology Project. Y.K. and Ms. Miwa Hirai are supported by Japan Agency for Medical Research and Development, Core Research for Evolutional Science and Technology (AMED-CREST), and establishment of the xenograft transplantation model using immunodeficient NOG mice was partly supported by a grant from Ajinomoto, Inc. The authors thank Ms. Miwa Hirai for her technical support of the manuscript preparation. We thank Yoshiko Naito, Tomomi Matsuura, Noriyo Hayakawa for their professional technical supports for metabolomics, and Professor Mayumi Kajimura and Dr. Ayako Yachie-Kinoshita for valuable discussion for manuscript preparation and systems biology in cancer, respectively.

## Author contributions

M.Sh. and M.Su. conceived and designed the whole study, and carried out data analyses and manuscript preparation. M.N. and S.Y. developed GNC and GNF substrates, respectively. M.Sh. and D.K. checked quality and plurality standards. T.T. examined simulation of electromagnetic fields based on morphological features of GNF substrates. H.T. contributed to morphological analyses of GNF substrates through scanning electron micrography. T.H. and A.K. contributes to identification and exclusion of many metabolites in the liver that can be visualized in GNF substrates. T.Y. and M.I. contributed to exploration of metabolic pathways for HT and taurine and carried out cell biology experiments. M.O. and Y.S and Y.K. designed and carried out experiments of a xenograft model of human colon cancer cell lines, and analyzed tissue sections for imaging MS. K.S., H.S. and O.N. supervised discussion on roles of CD44 in cancer survival and provided shCD44 HCT116 cells. H.N. and O.S. and T.M. performed experiments of brain tumour-initiating cells which were established by overexpressing H-RasV12 in normal neural stem/progenitor cells isolated from the SVZ of adult mice harbouring a homozygous deletion of the Ink4a/Arf locus. K.T. contributed to preparation of the manuscript. N.H. is a licensed professional clinical pathologist who examined T-bearing brain tissues. H.W. examined simulation analyses for molecular vibration of HT and glutathione.

## Additional information

**Competing interests:** The authors declare no competing interests.

