## [Peer Review File · Nature Communications]

Reviewers' comments:

Reviewer #1 (Remarks to the Author):

The authors use novel techniques to identify tumor-specific metabolites. Their invention of surface-enhanced Raman spectroscopy using gold nanofeave substrates appears to be a useful addition to the techniques that can be applied to study tumor metabolism in vivo.

1. My major concern is that most of the data (especially in vivo) and conclusions are derived from the analysis of a single cell line, HCT116. The authors should not present conclusions in general terms but refer specifically to the cell line(s) from which the data were derived.
2. The authors should become acquainted with the concept of significant figures. I cringe when I see biological data reported to four decimal places.
3. The authors provide two figures (Fig 8 and Fig 9) in which the conversion of 3-phosphoglycerate to serine is shown to be mediated by two enzymes, PHGDH and PSAT1. This is of course incorrect; it takes three enzymes to perform this reaction. Please include PSPH and show proper substrates and products of each of the three reactions.
4. The manuscript also requires careful editing by a native speaker of English to correct grammatical errors.
5. The authors are over-reaching in the last paragraph of the Discussion regarding clinical applications, particularly considering the data is primarily derived from a single colon carcinoma cell line.

Reviewer #2 (Remarks to the Author):

Shiota and her/his coworkers developed beautiful SERS model to characterize intrinsic Raman spectra. This work has high potential to be developed as a wide-field tip-enhanced Raman spectroscopy (TERS) and/or intrinsic Raman endoscope. This paper is also interesting as the authors demonstrated a Raman version of metabolic PET imaging, not only to distinguish tumor/normal tissue boundary via intrinsic Raman spectra, but to analyze metabolism of tumor/parenchymal tissue. This is a risky-as all the intrinsic Raman paper does, but a beautiful challenge. I recommend this article to publish in Nature communications after the revision.

- (1) Although they tried very hard to reach the solid discussion across the paper, the article is too long to publish in Nature communications. It is almost 6,000 words with 10 figures. Judging from the contents, there are lots of ways to polish the overall manuscript to be suitable for the publication.
- (2) Figure 1c, Figure 2a (in particular) should be magnified to clarify the morphology of the SERS structure.
- (3) Could you provide a fair comparison of the enhancement factor between the single nanofeave structure and nanosphere junction which is obtained from zero-angle metal deposition, which was also discussed in your previous paper, ACS Nano, 8, 5622-5632 (2014). In addition, what is the population of nanofeave junction rather than single nanofeave structure across the surface, which will definitely affect the overall enhancement factor of the film?
- (4) The nanofeave described here is so much similar to the nanorice, which is one of the best single particle SERS design for strong local electromagnetic enhancement. In this regard, please add the related reference of "Nanorice: A Hybrid Plasmonic Nanostructure. Nano Lett. 2006, 6, 827-832."
- (5) Is there a motivation for the authors to choose liver tumor model? Is that because there are more sufficient metabolites in liver than other organs so that you could build up intrinsic Raman libraries of metabolites easier?

- (6) In Fig. 3d, What is the 480 cm⁻¹ Raman shift responsible for?
- (7) When the Raman shift, 968 cm⁻¹ is considered as mainly hypotaurine (HT), could you provide a discussion why beta-carotene and retinol palmitate and retinol can't be also considered? It exhibits lower signal intensity, but not negligible.
- (8) I do love the MS analysis and DFT calculation, however, it would be better to make MS analysis discussion more concise.
- (9) Why there is an increase of GS and HT signal not only for shCD44, but also with shControl compared to the signals obtained from normal liver control (Fig.3 and 7)?

Reviewer #3 (Remarks to the Author):

This is a good but not excellent piece of work reporting on the use of large-area nanostructured substrates for surface-enhanced Raman screening detection of metabolites in tumor and normal tissues. It is based in the author's recent work in showing that bean-shaped gold nanostructures can be produced by angled deposition, and such shaped nanostructures show excellent properties for surface plasmonic enhancement. The current work is to apply this type of substrates for potentially distinguishing normal and tumor tissues by measuring their metabolites and other small molecules such as glutathione and hypotaurine. One major advantage is that this is a "label-free" approach, and the tissues can be measured without the use of injected agents or cellular/tissue stains.

The authors have carried out detailed and careful studies, but the overall approach has two major problems that will likely limit the utility of this strategy. First, metabolites and small molecules are often not a good indicator of tumor development or malignancy, so they are not reliable markers for differentiating tumors from healthy tissues. Second, there is considerable ambiguity because the reported spectral differences are not distinct enough for evaluating highly heterogeneous clinical tissue specimens. For these reasons, I don't believe that this work is suitable for publication in a Nature series journal.

Reply to Reviewers' comments:

General notes for the revised version of manuscript

(NCOMMS-17-21247A)

The authors thank 3 reviewers and Editorial Boards for giving us the opportunity to submit the revised version of our manuscript (NCOMMS-17-21247A).

The minor change of the title (~in cancer survival) resulted from correction by a native English speaker.

As reviewers #1 and #3 pointed out that conclusions of the original version (17-21247) are derived from the analyses of a single cell line, and that the data collected from xenograft transplantation of human colon cancer to immunodeficient mice hamper their feedback to apply SERS imaging to clinical cancer samples which display more complicated pathologic features. We were unable to accomplish a similar study using clinical cancer tissues, just because sufficient time to pass IRB paperworks within 3 months. However, in the revised version, we performed new sets of animal experiments to improve a quality of the study by using the murine model of *Ink4a/Arf*-null neural stem/progenitor cells expressing H-RasV12 and GFP (forming glioblastoma-like tumors with a 100% penetrance upon orthotopic implantation in syngeneic mice. The model enabled us to provide more complicated pathological features of glioblastoma with neovascularization, inflammatory responses and gliosis in the surrounding brain tissues. (Saga I, Shibao S, Okubo J, et al. Integrated analysis identifies different metabolic signatures for tumor-initiating cells in a murine glioblastoma model. *Neuro Oncol.* 2014;16(8):1048–1056.)

These data were newly added in Fig. 10 and Suppl. Fig. 15 in the revised version (17-21247A). In this model, distribution of GFP-positive cancer cells does not necessarily exhibit “solid” shapes but display anatomically amorphous main tumour with multiple daughter micro-tumours surrounding the main one that coincide with infiltration of inflammatory cells and/or gliosis. During the limited time for revision. Using this model, we were able to build up “accumulative lists of Raman shifts (ALR in Fig. 4 and Suppl. Fig. 3)” and a single mouse was used for practicing automated extraction of the tumour boundaries based on the ALR-database. Since this model displayed complicated tumour boundaries including irregular borders with reactive gliosis and inflammatory responses surrounding the main tumour, we asked a national-licensed experienced pathologist to carefully extract the tumour regions that did not coincided with gliosis and inflammatory responses to build up the database of T-dominant and P-dominant Raman shifts”. As seen in Fig. 10 and Table S3, SERS imaging and region-specific analyses of the spectral features clearly showed that the

technology enables us to clearly distinguish the tumour regions. Such novel data showed that GNF-SERS technology is applicable to the extraction of tumour boundaries even in the regions displaying more complicated features of malignancy, and thus serve as a reply to the Reviewer 3. At the same time, according to the reviewers' and Editorials' suggestions, we softened overarching description of the clinical application of our GNF-SERS technology in the last portion of Discussion.

Immediately after receiving the editorial comments on October 23, 2017, we did our best to submit a proposal for clinical research in which we attempted to use surgical post-operative human samples of carcinoma to apply our SERS technology under collaboration with National Cancer Center Hospital in Japan. While we are still waiting for the answer from IRB of the Institute, it seems impossible to have a permission before the date of the deadline (January 23, 2018).

I appreciated if the Editorial Boards understand our difficulty to perform clinical studies using human pathological samples which usually require much time to be approved. Instead, what we did the best for revision within 3 months is to provide the new datasets of the current mouse glioblastoma model, by which we believe usefulness of the current GNF-SERS technology has been empowered.

According to the Reviewer 2's suggestion, we transferred the major portions of mass spectroscopic data into Suppl. Information. These portions were just transferred to Suppl. Information without changing the sentences that were underlined with broken lines. On the other hand, all revised portions which were newly added in the revised version were marked by "solid underlines" in the revised manuscript. Suppl. Fig. 2, Suppl. Fig. 4b, Fig. 10 and Suppl. Fig. 15 constitute major portions that were newly added according to the Reviewers' suggestions.

Because of considerable contribution to accomplishment of new data from in vivo tumours with a perfect penetrance on orthotopic implantation of cancer stem cells in syngenic mice, I would like to add the following co-authors: Dr. Oltea Sampetean (oltea@a6.keio.jp) and Dr. Hideaki Nagashima (032m2068@gmail.com) who developed and performed important experiments of the syngenic cancer stem cell model of glioblastoma, and Dr. Nobuyoshi Hiraoka (nhiraoka@ncc.go.jp) who is a professional pathologist who evaluated pathology of mouse glioblastoma-bearing brain tissues. In addition, Dr. Takayuki Morikawa (morikawa150401@gmail.com) contributed to preparation of frozen brain tissues, and Dr. Keiyo Takubo (keiyot@gmail.com) contributed to preparation of the revised manuscript as an expert of stem cell biology. All other authors agreed with adding these important contributors in the revised version.

We asked a professional English writer to correct English. Such corrections were not underlined. Otherwise, all revised portions were underlined in the text, so that the Referees can easily recognize all revised portions of the manuscript.

Reviewer #1 (Remarks to the Author):

Q1: *The authors use novel techniques to identify tumor-specific metabolites. Their invention of surface-enhanced Raman spectroscopy using gold nanofe substrates appears to be a useful addition to the techniques that can be applied to study tumor metabolism in vivo.*

1. My major concern is that most of the data (especially in vivo) and conclusions are derived from the analysis of a single cell line, HCT116. The authors should not present conclusions in general terms but refer specifically to the cell line(s) from which the data were derived.

A1: Thank you for pointing out a critical weakness of the original manuscript. Since we believe that the point raised by Reviewer #1 is important, we added new sets of experiments using the model of murine *Ink4a/Arf*-null neural stem/progenitor cells expressing H-RasV12 and GFP: The model forms glioblastoma-like tumors with a 100% penetrance upon orthotopic implantation in syngeneic mice, and displays complicated pathological features characterized by gliosis, neovascularization and inflammatory responses. (Saga I, Shibao S, Okubo J, et al. Integrated analysis identifies different metabolic signatures for tumor-initiating cells in a murine glioblastoma model. *Neuro Oncol.* 2014;16(8):1048–1056.) Results were shown in Fig. 10 and Suppl. Fig. 15. The magnitude of GFP expression highly depends on H-ras expression. We therefore asked professional clinical pathologist (Dr. Nobuyoshi Hiraoka as a new co-author) to carefully examine pathological tissue specimens of the glioblastoma model to extract regions of tumours, those showing tumour-associated responses, and tumour-free intact brain tissues. Because of the higher complexity of pathological features, we compared tumour-bearing ipsilateral hemisphere with tumour-free contralateral hemisphere of the cancer stem cell-implanted brain, or control mice which are not implanted with the cancer stem cells. As seen in Fig. 10 in the revised version, difference spectrum between tumour-bearing and tumour-free regions displayed tumour-specific and parenchyma (striatum)-specific Raman peaks. Furthermore, the image processing by the algorithm shown in the original version (Fig 4 and Suppl Fig 3 in the revised version: no changes) allowed us to extract the localization of glioblastoma (Figs 10e-f). These lines of data also appear to cover an important portion of the answer to a question raised by the Referees #1 and #3 who asked us to examine if the current system is applicable to other models of malignancy with highly complicated pathological features (Page 11 and Fig 10 and Suppl Fig 15 in the revised version).

The authors agreed with the Reviewer's criticism that *most of the data (especially in vivo) and conclusions are derived from the analysis of a single cell line, HCT116*. Therefore, adding these new datasets, we rephrased the last paragraph of Discussion as follows: Detection of global metabolic profiles by GNF-SERS benefits understanding robustness of

metabolic systems of cancer⁶⁰, at least in the two different models of cancer development in mice. However, under clinical settings, post-operative cancer specimens undergo an exposure to vascular ligation that might largely alter metabolomics of samples. Thus, further studies using clinical specimens are obviously necessary to determine whether SERS imaging is applicable to pathological diagnosis of cancer (Page 12).

Q2. *The authors should become acquainted with the concept of significant figures. I cringe when I see biological data reported to four decimal places.*

A2: Thank you for noticing us problematic presentation for data, which has been corrected through the revised version

Q3. *The authors provide two figures (Fig 8 and Fig 9) in which the conversion of 3-phosphoglycerate to serine is shown to be mediated by two enzymes, PHGDH and PSAT1. This is of course incorrect; it takes three enzymes to perform this reaction. Please include PSPH and show proper substrates and products of each of the three reactions.*

A3: Thank you indeed for raising this question that gave us the opportunity to provide an important information on how C1 unit is supplied to synthesize serine and hypotaurine/taurine as final products. In the revised version, we corrected the reaction pathway so that PHGDH, PSAT1 and PSPH appear in Figs. 7 and 8 in the revised version (Pages 35-36). Furthermore, the authors greatly thank this Reviewer's question, because we performed additional experiments to visualize PSPH protein expression using the same lysate samples used in the original version: As seen in Fig. 8e (Page 35) in the revised version, protein expression of PHDGH, PSAT1 and PSPH was all increased in response to CD44 knockdown in HCT116 cells. According to these new results, a portion of abstract and methods were altered and underlined in the revised version.

Furthermore, using the same lysate samples for CE-MS metabolomics, we determined phospho-serine (P-serine) and its mass-labeled derivatives: As seen in Fig. 7 in the revised version, the newly added data clearly indicated that ¹³C₃-labeled 3-PG in the glycolytic pathway is converted to ¹³C₃-P-serine with statistical significance, while (as shown in the original manuscript) serine is detectable mainly as ¹³C₁- and ¹³C₃-labeled serine with statistical significance versus ¹³C₂-serine. These results strengthen a hypothesis that the C₁-group transfer mechanism plays a role in generating ¹³C₁-serine as a source of hypotaurine/taurine in HCT116 cells. The discussion was briefly added in the revised version (Figs 7 and 8 were revised according to the Reviewer 1). PSPH protein expression and the data of p-serine was inserted in the text with being underlined.

Q4. *The manuscript also requires careful editing by a native speaker of English to correct grammatical errors.*

A4: Thank you indeed for pointing it out. The revised manuscript has been corrected by professional a native English speaker with knowledge and experiences of life sciences.

Q5. *The authors are over-reaching in the last paragraph of the Discussion regarding clinical applications, particularly considering the data is primarily derived from a single colon carcinoma cell line.*

A5: Thank you for raising an important comment. As we replied in A1, The final paragraph of Discussion was rephrased to soften the expression of clinical application, since we really need to refine the technology and protocols for future studies using clinical samples. As pointed out by the Reviewer, the original version showed the data primarily derived from a single cancer cell line that causes simple morphological features of metastases, we added new sets of experiments using the model of murine *Ink4a/Arf*-null neural stem/progenitor cells expressing H-RasV12 and GFP which forms glioblastoma-like tumors. This model mimics development of human glioblastoma with a 100% penetrance upon orthotopic implantation in syngeneic mice. In this model, pathology indicated very complicated mixture of morphologically irregular tumour bodies with gliosis and inflammatory responses. The results showed that SERS imaging not only recognized xenograft tumour metastases in the liver, but also determined tumour regions with gliosis and infiltration of inflammatory cells, and those of non-tumour regions in brain, suggesting that the technology benefit diagnosis even for the rather complicated glioblastoma than xenograft colon cancer cell transplantation (Page 12 and Fig 10).

Regarding Discussion that included the application to human pathology, we deleted overarching description in the original version, and clarified unresolved problems to challenge for clinical application using human samples (page 12, See our reply A1).

Reviewer #2 (Remarks to the Author):

Shiota and her coworkers developed beautiful SERS model to characterize intrinsic Raman spectra. This work has high potential to be developed as a wide-field tip-enhanced Raman spectroscopy (TERS) and/or intrinsic Raman endoscope. This paper is also interesting as the authors demonstrated a Raman version of metabolic PET imaging, not only to distinguish tumor/normal tissue boundary via intrinsic Raman spectra, but to analyze metabolism of

tumor/parenchymal tissue. This is a risky-as all the intrinsic Raman paper does, but a beautiful challenge. I recommend this article to publish in Nature communications after the revision.

Q1: *Although they tried very hard to reach the solid discussion across the paper, the article is too long to publish in Nature communications. It is almost 6,000 words with 10 figures. Judging from the contents, there are lots of ways to polish the overall manuscript to be suitable for the publication.*

A1: Thank you for your suggestion to shorten the manuscript. In the revised version, we prepared the main text (Intro~Discussion) within 5000 words (4991 words) by moving considerable text portions and the related figures showing the data of mass spectrometry towards Suppl. Info. These lines of information remained unchanged in the revised version, and underlined by broken lines in Suppl. Info. Our answer to Q1 is also consistent with a reply to Question 8 from the Reviewer 2.

Q2: *Figure 1c, Figure 2a (in particular) should be magnified to clarify the morphology of the SERS structure.*

A2: We newly added experiments to collect high magnification electron micrographs that I hope benefit characterization of a shape of single Au-nanoparticles. These new SEM images were installed in Fig. 1d and Fig. 2a.

Q3: *Could you provide a fair comparison of the enhancement factor between the single nanofeve structure and nanosphere junction which is obtained from zero-angle metal deposition, which was also discussed in your previous paper, ACS Nano, 8, 5622-5632 (2014). In addition, what is the population of nanofeve junction rather than single nanofeve structure across the surface, which will definitely affect the overall enhancement factor of the film?*

A3: Thank you very much for an intriguing question. According to the Referee's suggestion, we attempted to provide a fair composition of the enhancement factor (G values in Fig. 2 and Suppl. Fig. 2) between single nanofeve structures and nanosphere junction. The density of Au-nanoparticles corresponds to the "gap distance" between the two adjacent nanoparticles. Therefore, in the revised version, we examined differences in the G values as a function of the gap distance (also called "junction" in the other literature including Ref. 20 cited in the revised version) between GNF and GNC. As seen in Suppl. Fig. 2, FDTD simulation revealed that the local field enhancement by two adjacent Au-nanoparticles of

GNC are diminished by increasing the gap distance over 10 nm (typical gap distance of GNF in **Fig.2a**), suggesting that the density of GNC nano-particles rate-limits the enhancement. In contrast, the enhancement by GNF nanoparticles does not regress and stay in a plateau level so far in a distance less than 200 nm of the gap distance (**Suppl. Figs. 2a-c**). As shown in a high-magnification SEM (Fig. 1d in the revised version, the approximate distance between the adjacent Au-nanoparticles in GNF is in a range between 50~100 nm. Collectively, these results suggest that the anisotropy of Au-nanoparticles rather than effects of the gap of the Au-nanoparticles, which is also called “junction”, determine SERS enhancement in GNF substrates, being in good agreement with previous studies (Ref. 20 cited in the revised version).

Q4: *The nanofeve described here is so much similar to the nanorice, which is one of the best single particle SERS design for strong local electromagnetic enhancement. In this regard, please add the related reference of “Nanorice: A Hybrid Plasmonic Nanostructure. Nano Lett. 2006, 6, 827-832.”*

A4: The authors thank this question from Reviewer #2. We added brief comments with citing the reference of this intriguing publication (Reference #20 in the revised version). Eventually, the citation of this article with FDTD simulation to compare GNC and GNF allowed us to emphasize the importance of morphologic anisotropy than the gap distance (“junction”) of Au-nanoparticles as mechanisms for enhancing the electromagnetic fields (Suppl. Fig. 2 in the revised version).

Q5. *Is there a motivation for the authors to choose liver tumor model? Is that because there are more sufficient metabolites in liver than other organs so that you could build up intrinsic Raman libraries of metabolites easier?*

A5. Thank you for an excellent question. There were several motivations to use a colon cancer xenograft model in the original version. (1) high reproducibility of metastases, (2) Likewise clinical cases of liver metastases of colon cancer, the model provided similar phenotypes of cancer metastasis with relatively clear borders between tumours and parenchyma. (3) Data availability of quantitative tissue contents of individual metabolites in the liver that were collected either by metabolomics or from previous publications in our laboratory and others. This is important to establish libraries of SERS signals which were derived from physiologically relevant concentrations of metabolites.

However, in the revised version, we attempted to examine if SERS imaging is useful to extract the borders between tumours and parenchyma in other organs such as brain.

While the addition of these datasets was based on a criticism raised by other referees, this question raised by the Reviewer 2 led us to examine whether the algorithms developed in this study (Fig. 4 and Suppl. Fig. 3) is useful to extract the boundaries of tumours which are developing in other organs such as brain. To collect such data on T-dominant and P-dominant fingerprints of SERS signals, we applied the model of murine *Ink4a/Arf*-null neural stem/progenitor cells expressing H-RasV12 and GFP which forms glioblastoma-like tumors. The difference spectra between the ipsilateral glioma regions and the contralateral normal regions (striatum, Suppl. Fig 15a and Fig 10 in the revised version) indicated T-dominant and P-dominant SERS peaks, suggesting the presence of metabolic fingerprints to discriminate the two regions.

To extract the boundaries between tumours and parenchyma, the data of Raman peaks were not accumulated, but processed directly at the smallest detector resolution ($\sim 2 \text{ cm}^{-1}$). The protocol of this kind benefits precise extraction of the specific Raman shifts **(Suppl. Tab. S1)** to distinguish tumour-dominant signals from parenchyma-dominant ones. This process was necessary to improve the accuracy rates of tumour boundary extraction as described later in this section. As a result, GNF-SERS analyses yielded 16 T-dominant and 10 P-dominant peaks (Suppl. Fig. 15a and Table S3) in the revised version. Furthermore, The GNF-SERS imaging for accumulative data (central peaks $\pm 10 \text{ cm}^{-1}$) revealed the two major Raman peaks at 485 cm^{-1} and 726 cm^{-1} . The peak at 726 cm^{-1} appeared to be derived from metabolites with adenine ring (e.g. IMP and AMP. See Reference 15 from our lab.). Such a hypothesis was in good agreement with the vacuum-type imaging MS data showing distribution of AMP and IMP in the serial tissue section (Fig. 10g in the revised version).

As the reviewer mentioned, livers seem to possess larger number of P-dominant marker peaks than brain. Furthermore, in case of brain analyses, we must further consider intra-organ spatial heterogeneity of metabolites. Because of this, the striatum region of the contralateral hemisphere was carefully chosen for the control region versus glioblastoma occurring in striatum of the ipsilateral hemisphere. These lines of the data were added in the revised version (Page 10, last para. with Fig 10 and Suppl Fig 15). Thank you for your question on this important issue.

Q6 In Fig. 3d, what is the 480 cm^{-1} Raman shift responsible for?

A6. Thank you for asking us the very important question. In the original version of our manuscript, we were unable to identify molecules responsible for the SERS peak at 480 cm^{-1} . The question from the other reviewers led us to perform additional experiments using an orthotopic syngeneic mouse model of glioblastoma, and the analyses of difference GNF-SERS spectra (Suppl. Fig 15a) gave us a clue to answer to this question ! Interestingly,

likewise in the liver xenograft cancer model, we observed the SERS peak at 485 cm^{-1} as a T-dominant peak (Suppl Fig 15a). Careful surveillance of previous Raman reports (Janz GJM et al. 1976 Inorg Chem, reference 47) showed that S-S stretches of polysulfide compounds yield the peaks at around $460\sim 480\text{ cm}^{-1}$. Based on the conventional analyses (Ref. 47) of crystals of Na_2S_x compounds which are commercially available, we collected the new data showing Na_2S_3 and Na_2S_4 yields robust Raman signals with multiple and broad peaks at around $450\sim 485\text{ cm}^{-1}$, which coincided with the peak at $< 300\text{ cm}^{-1}$, indicating the Au-S stretch (Suppl Fig 15c). Use of GNF-SERS for examining these sodium polysulfides in distilled water (Suppl Fig 15d) markedly enhanced both Au-S and S-S stretch signals. Although the concentrations of Na_2S_x , which were used in the experiments in vitro, seemed high as compared with the physiological concentrations (Few methods have been available to determine the exact contents of these metabolites), these results raised a possibility that endogenous polysulfide compounds serve as candidates of metabolites yielding the peak at $480\sim 485\text{ cm}^{-1}$.

While there are only limited methods to determine specific polysulfide compounds, overall amounts of H_2S (the compound yielding no peak at 480 cm^{-1} : Suppl Fig 4a) and polysulfides were able to be determined by monobromobimane-assisted derivatization of S-containing compounds (References 39 and 48) using LC-MSMS. We thus conducted new experiments to measure these metabolites, and the result showed that the tumour-bearing brain tissues displayed significant elevation of H_2S /polysulfides than the tumour-free brain tissues (Suppl Fig 15e). This result is consistent with a notion that glioblastoma generates endogenous polysulfides, which result in generation of T-dominant SERS peaks at around 480 cm^{-1} . Fig. 3b indicated the peaks at 480 cm^{-1} occurs in colon cancer xenografts as well as the liver parenchyma. Since the liver constitutes another source of endogenous polysulfides, according to previous articles showed (References 52 and 67), the presence of SERS peaks in the liver (and tumours) at 480 cm^{-1} does not seem unreasonable.

Q7 : When the Raman shift, 968 cm^{-1} is considered as mainly hypotaurine (HT), could you provide a discussion why beta-carotene and retinol palmitate and retinol can't be also considered? It exhibits lower signal intensity, but not negligible.

A7: Thank you for your careful review and comments. The Reviewer 2's question led us to re-examine the "optimal" peak for hypotaurine which includes the shoulder of retinol/retinol palmitate and beta-carotene at 968 cm^{-1} . First, we measured amounts of retinol palmitate, retinol and beta-carotene in the control mouse liver (Suppl. Info. Page 3, 1st para). The data showed that retinol palmitate constituted a major metabolite, the concentration of which was $\sim 0.9\text{ mM}$, which retinol and beta-carotene were undetectable. Thus, in new experiments in vitro, we compared SERS signals of hypotaurine and retinol

palmitate at 1 mmol/L in methanol solution (Suppl. Fig 4b). By shifting the summation band to higher wave number towards 978 cm^{-1} , we were able to extract HT signal with minimizing contamination of signals derived from retinol palmitate. Accordingly, HT data were reanalyzed by a new criterion of the summation of the band for wave numbers ($978 \pm 8\text{ cm}^{-1}$ in Suppl. Figs. 4b-c in the revised version). The description of the corrected wave number was underlined throughout the text and Figs and Suppl. Info.

Q8. *I do love the MS analysis and DFT calculation, however, it would be better to make MS analysis discussion more concise.*

A8. Thank you for your question: in the revised version, we moved most of the datasets of MS analyses towards Suppl. Info. All portions which were moved to Suppl. Information were identical to those in the original version, and underlined by broken lines.

Q9 *Why there is an increase of GS and HT signal not only for shCD44, but also with shControl compared to the signals obtained from normal liver control (Fig.3 and 7)?*

A9. Thank you for an important question. According to the Reviewer's suggestion in Q7, we re-assessed the optimal SERS band for HT at 978 cm^{-1} (Fig. 7). Under these circumstances, there were no differences of HT signals among the normal, shControl- and shCD44-bearing livers.

As the Reviewer pointed out, the GS signal increased in tumour-bearing livers (both parenchyma (P) and tumours (T)). We previously reported that GSH elevation results from CD44-mediated stimulation of cysteine incorporation into cancer cells (Reference 11 in the revised version). While detailed mechanisms were unknown, we previously reported that tumour metastases caused regenerative responses of the liver parenchyma which secondarily stimulate GSH elevation in the parenchyma (Reference 12). These lines of description were briefly added in the revised version.

Reviewer #3 (Remarks to the Author):

This is a good but not excellent piece of work reporting on the use of large-area nanostructured substrates for surface-enhanced Raman screening detection of metabolites in tumor and normal tissues. It is based in the author's recent work in showing that bean-shaped gold nanostructures can be produced by angled deposition, and such shaped nanostructures show excellent properties for surface plasmonic enhancement. The current

work is to apply this type of substrates for potentially distinguishing normal and tumor tissues by measuring their metabolites and other small molecules such as glutathione and hypotaurine. One major advantage is that this is a "label-free" approach, and the tissues can be measured without the use of injected agents or cellular/tissue stains. The authors have carried out detailed and careful studies, but the overall approach has two major problems that will likely limit the utility of this strategy. First, metabolites and small molecules are often not a good indicator of tumor development or malignancy, so they are not reliable markers for differentiating tumors from healthy tissues. Second, there is considerable ambiguity because the reported spectral differences are not distinct enough for evaluating highly heterogeneous clinical tissue specimens. For these reasons, I don't believe that this work is suitable for publication in a Nature series journal.

Thank you for giving us critical comments. As the Reviewer pointed out, small molecules are often not reliable markers for differentiating tumours from healthy tissues. In the revised version, we therefore deleted a description in Discussion which suggested clinical application of the technique, but added some problems (e.g. effects of vascular ligation on alterations in small molecular metabolites in tumours and surrounding tissues), when the technique is applied for clinical samples. But, on the other hand, we would emphasize that the tumour borders can be extracted by examining 28 (6 are Tumour-dominant and 22 are Parenchyma-dominant. See Table S1 in the original version) SERS peaks, but not by determining only 2 metabolites such as GS and HT.

Another point raised by the Referee is "considerable ambiguity", because the reported spectral differences are not distinct enough for evaluating highly heterogeneous clinical tissue specimens. We attempted to tackle this problem by using clinical samples and submitted protocols of clinical study to institutional IRB, but unfortunately, it seemed to take more than 3 months because of a slow process of paper works. Instead, to overcome this problem, we performed new animal experiments in the revised version using the model of murine *Ink4a/Arf*-null neural stem/progenitor cells expressing H-RasV12 and GFP (forming glioblastoma-like tumors with a 100% penetrance upon orthotopic implantation in syngeneic mice. The glioblastoma model enabled us to examine more complex pathological features analogous to human glioblastoma with vascular involvement, inflammatory responses and gliosis in the surrounding brain tissues (Refs 45~47) SERS imaging and region-specific analyses of the spectral features revealed several tumour-dominant and parenchyma-dominant metabolites, and the current method enabled us to distinguish the tumour regions from non-tumour intact regions even in this orthotopic syngeneic mouse glioblastoma model which showed complicated pathology. These data were added in Fig. 10 and Suppl. Fig. 15 in the revised version (17-21247A).

REVIEWERS' COMMENTS:

Reviewer #1 (Remarks to the Author):

The authors have revised the manuscript to address concerns from all of the reviewers, including the analysis of a second tumor model. The manuscript contains an impressive amount of data. In the Discussion, the authors did a good job of balancing the novelty and utility of their technique with a discussion of its potential limitations. The work will be of great interest to investigators of cancer metabolism.

Reviewer #2 (Remarks to the Author):

The authors revised the manuscript extensively from the discussion points that reviewers have concerned. In particular, I am pleased to see the glioblastoma (GBM) intrinsic Raman image in addition to the tumor in liver. I have the last additional request before publication. Because there is GFP expression in your GBM model, please discuss the supporting proof that intrinsic Raman from GFP doesn't affect your conclusion of the margination between tumor and normal tissue. If the question is well-addressed, I recommend the publication to be published in Nature Communications.

Reviewer #3 (Remarks to the Author):

The authors have made a good effort to address my previous concerns, although it is still too early to determine if the SERS method is sufficiently reliable or reproducible for future clinical applications. Overall, the revised manuscript has been improved with more experimental data, and a number of important issues have been clarified or explained. So I believe that this work is now suitable for publication in Nature Communications.

Point-by-point reply to Reviewers' comments:

Reviewer #1 (Remarks to the Author):

The authors have revised the manuscript to address concerns from all of the reviewers, including the analysis of a second tumor model. The manuscript contains an impressive amount of data. In the Discussion, the authors did a good job of balancing the novelty and utility of their technique with a discussion of its potential limitations. The work will be of great interest to investigators of cancer metabolism.

Answer to Reviewer #1: Thank you very much for your positive comments. We were delighted to have careful advises from Reviewer 1.

Reviewer #2 (Remarks to the Author):

The authors revised the manuscript extensively from the discussion points that reviewers have concerned. In particular, I am pleased to see the glioblastoma (GBM) intrinsic Raman image in addition to the tumor in liver. I have the last additional request before publication. Because there is GFP expression in your GBM model, please discuss the supporting proof that intrinsic Raman from GFP doesn't affect your conclusion of the margination between tumor and normal tissue. If the question is well-addressed, I recommend the publication to be published in Nature Communications.

Answer to Reviewer #2: Thank you for a valuable question raised by Reviewer #2. In the revised version NCOMMS-17-21247B, we added new experiments to examine whether GFP yields GNFSERS signals at a concentration which is comparable to that in tumour-bearing brain. In the last paragraph of Page 20 in the Main Text file, we described methods briefly, and showed the results indicating that GFP at supraphysiological density yields no SERS signals in vitro as indicated in Supplementary Figure 4a. The results exclude a possibility that SERS signals derived from GFP in brain tumours contributes to extraction of the cancer localization in tissues.

Reviewer #3 (Remarks to the Author):

The authors have made a good effort to address my previous concerns, although it is still too early to determine if the SERS method is sufficiently reliable or reproducible for future clinical applications. Overall, the revised manuscript has been improved with more experimental data, and a number of important issues have been clarified or explained. So I believe that this work is now suitable for publication in Nature Communications.

Answer to Reviewer #3: Thank you very much for your positive comments. We were delighted to have careful and fruitful advises from the Reviewer 3 which led us to add neuroblastoma experiments.